# On the Adversarial Robustness of Vision Transformers

**Rulin Shao**                                                              *rulins@cs.cmu.edu*
*Carnegie Mellon University*

**Zhouxing Shi**                                                              *zshi@cs.ucla.edu*
*University of California, Los Angeles*

**Jinfeng Yi**                                                              *yijinfeng@jd.com*
*JD AI Research*

**Pin-Yu Chen**                                                              *pin-yu.chen@ibm.com*
*IBM Research*

**Cho-Jui Hsieh**                                                              *chohsieh@cs.ucla.edu*
*University of California, Los Angeles*

**Reviewed on OpenReview:** *https://openreview.net/forum?id=1E7K4n1Esk*

## Abstract

Following the success in advancing natural language processing and understanding, transformers are expected to bring revolutionary changes to computer vision. This work provides a comprehensive study on the robustness of vision transformers (ViTs) against adversarial perturbations. Tested on various white-box and transfer attack settings, we find that ViTs possess better adversarial robustness when compared with MLP-Mixer and convolutional neural networks (CNNs) including ConvNeXt, and this observation also holds for certified robustness. Through frequency analysis and feature visualization, we summarize the following main observations contributing to the improved robustness of ViTs: 1) Features learned by ViTs contain less high-frequency patterns that have spurious correlation, which helps explain why ViTs are less sensitive to high-frequency perturbations than CNNs and MLP-Mixer, and there is a high correlation between how much the model learns high-frequency features and its robustness against different frequency-based perturbations. 2) Introducing convolutional or tokens-to-token blocks for learning high-frequency features in ViTs can improve classification accuracy but at the cost of adversarial robustness. 3) Modern CNN designs that borrow techniques from ViTs including activation function, layer norm, larger kernel size to imitate the global attention, and patchify the images as inputs, etc., could help bridge the performance gap between ViTs and CNNs not only in terms of performance, but also certified and empirical adversarial robustness. Moreover, we show adversarial training is also applicable to ViT for training robust models, and sharpness-aware minimization can also help improve robustness, while pre-training with clean images on larger datasets does not significantly improve adversarial robustness. Codes available at `https://github.com/RulinShao/on-the-adversarial-robustness-of-visual-transformer`.

## 1 Introduction

Transformers are originally applied in natural language processing (NLP) tasks as a type of deep neural network (DNN) mainly based on the self-attention mechanism (Vaswani et al., 2017; Devlin et al., 2018; Brown et al., 2020), and transformers with large-scale pre-training have achieved state-of-the-art results on many NLP tasks (Devlin et al., 2018; Liu et al., 2019; Yang et al., 2019; Sun et al., 2019). Recently, Dosovitskiy et al. (2020) applied a pure transformer directly to sequences of image patches (i.e., a vision transformer, ViT) and showed that the Transformer itself can be competitive with convolutional neural networks (CNN) on image classification tasks. Since then transformers

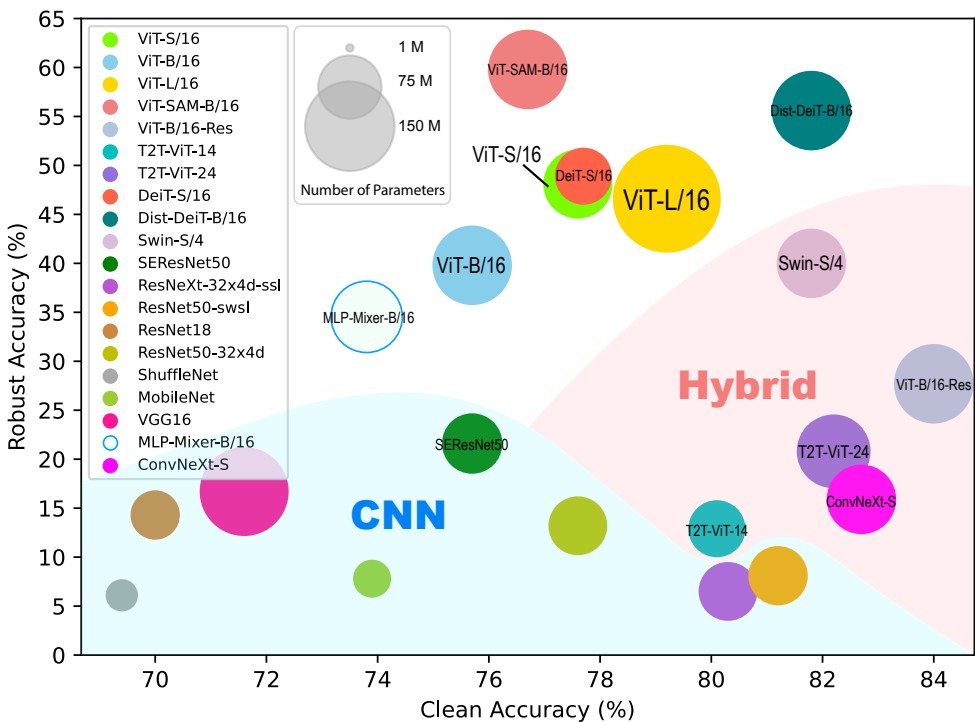

Figure 1: Robust accuracy v.s. clean accuracy. The robust accuracy is evaluated by AutoAttack (Croce & Hein, 2020a). The "Hybrid" class includes CNN-ViT, T2T-ViT and Swin-T as introduced in Section 3. Models with attention mechanisms have their names printed at the center of the circles. ViTs have the best robustness against adversarial perturbations. Introducing other modules to ViT can improve clean accuracy but hurt adversarial robustness. CNNs are more vulnerable to adversarial attacks.

have been extended to various vision tasks and show competitive or even better performance compared to CNNs and recurrent neural networks (RNNs) (Carion et al., 2020; Chen et al., 2020; Zhu et al., 2020). While ViT and its variants hold promise toward a unified machine learning paradigm and architecture applicable to different data modalities, it is critical to study the robustness of ViT against adversarial perturbations for safe and reliable deployment of many real-world applications.

In this work, we examine the adversarial robustness of ViTs on image classification tasks and make comparisons with CNN and MLP baselines. As highlighted in Figure 1, our experimental results illustrate the superior robustness of ViTs than CNNs and MLP-Mixer in various settings, based on which we make the following important findings:

- Features learned by ViTs contain less high-frequency information and benefit adversarial robustness. ViTs achieve a higher robust accuracy (RA) of $59.8\%$ compared with a maximum improvement of $16.7\%$ by CNNs in Figure 1. They are also less sensitive to high-frequency adversarial perturbations.

- Using denoised randomized smoothing (Salman et al., 2020), ViTs attain significantly better certified robustness than CNNs.

- It takes the cost of adversarial robustness to improve the classification accuracy of ViTs by introducing blocks to help learn low-level features as shown in Figure 1.

- Increasing the proportion of transformer blocks in the model leads to better robustness when the model consists of both transformer and CNN blocks. For example, the robust accuracy (RA) increases from $12.9\%$ to $20.8\%$ when 10 additional transformer blocks are added to T2T-ViT-14. However, increasing the size of a pure transformer model cannot guarantee a similar effect, e.g., the robustness of ViT-S/16 is better than that of ViT-B/16 in Figure 1.

- The principle of adversarial training through min-max optimization (Madry et al., 2017; Zhang et al., 2019) can be applied to train robust ViTs. Pre-training on larger datasets without adversarial training does not improve adversarial robustness. But sharpness-aware optimization (Foret et al., 2020; Chen et al., 2021) benefits both the adversarial robustness and clean accuracy of ViTs.

## 2 Related Work

Transformer (Vaswani et al., 2017) has achieved remarkable performance on many NLP tasks, and its robustness has been studied on those NLP tasks. Hsieh et al. (2019); Jin et al. (2020); Shi & Huang (2020); Li et al. (2020); Garg & Ramakrishnan (2020); Yin et al. (2020) conducted adversarial attacks on transformers including pre-trained models, and in their experiments transformers usually show better robustness compared to other models based on Long short-term memory (LSTM) or CNN, with a theoretical explanation provided in Hsieh et al. (2019). However, due to the discrete nature of NLP models, these studies are focusing on discrete perturbations (e.g., word or character substitutions) which are very different from small and continuous perturbations in computer vision tasks. Besides, Wang et al. (2020a) improved the robustness of pre-trained transformers from an information-theoretic perspective, and Shi et al. (2020); Ye et al. (2020); Xu et al. (2020) studied the robustness certification of transformer-based models. Recently, there are some concurrent works studying the adversarial robustness of ViTs. In the context of computer vision, one earliest relevant work is Alamri et al. (2020) which applies transformer encoder in the object detection task and reports better adversarial robustness. But their considered model is a mix of CNN and transformer instead of the ViT model considered in this paper. Besides, the attacks they applied were relatively weak, and there lacks studies and explanations on the benefit of adversarial robustness brought by the transformers.

Recently, there are many concurrent or follow-up works investigating the adversarial robustness of vision transformers. Many of them have cited a preprint version of our work and shown valuable discussion or extended experiments. We acknowledge their contributions below. Mahmood et al. (2021) test the adversarial robustness of the transformer in both white-box and black-box settings, analyze the security of a simple ensemble defense of CNNs and transformers, and find such ensemble defense is effective for the black-box setting. Qin et al. (2021) and Salman et al. (2021) investigate the adversarial robustness of ViTs through the lens of patch-based architectural structure. Herrmann et al. (2022) further designs a pyramid adversarial training with augmentation techniques to improve the ViT's both sanity and robust performance. Naseer et al. (2021b) investigate whether the weak transferability of adversarial patterns from high-performing ViT models, as reported in our work, is a result of weak features or a weak attack. Aldahdooh et al. (2021); Naseer et al. (2021a); Paul & Chen (2021); Tang et al. (2021); Mao et al. (2022) study the adversarial robustness from different views, e.g., preprocessing defense methods (Aldahdooh et al., 2021), shape recognition capability (Naseer et al., 2021a), natural adversarial examples (Paul & Chen, 2021; Tang et al., 2021), and detailed robust components in ViTs (Mao et al., 2022). (Jeeveswaran. et al., 2022) studies object detection and semantic segmentation tasks, and shows ViTs are more robust to distribution shifts, natural corruptions, and adversarial attacks in both tasks.

Our work differs from these concurrent and related works by focusing more on the origin of the adversarial robustness of vision transformers. We discuss the superior adversarial robustness of ViTs through the lens of frequency, and find that the ViTs are especially robust to high-frequency adversarial perturbations. We also apply denoised randomized smoothing and show ViT also has superior certified robustness than CNN models. Our study provides insight on understanding the source of ViT's adversarial robustness and designing more robust architectures. And we cover various baselines in the experiments for comprehensive comparison, including the recently proposed ConvNeXt (Liu et al., 2022), MLP-Mixer (Tolstikhin et al., 2021), and Swin-Transformers (Liu et al., 2021), etc.

To the best of our knowledge, our work is the first study that investigates the certified adversarial robustness of transformers on computer vision tasks, and explains the source of ViTs' superior adversarial robustness from a frequency perspective. We also show that ViTs have superior adversarial robustness (against small perturbations in the input pixel space) in both white-box and black-box settings. And we investigate various SOTA models including the recently proposed ConvNeXt (Liu et al., 2022) and MLP-Mixer (Tolstikhin et al., 2021) in the experiments.

# 3 Model Architectures

We first review the architectures of models investigated in our experiments. A summary of the target models investigated in the main text is shown in Table 1. More experimental results of ViT variants can be found in Appendix B. The weights of all the investigated models are all **publicly available** checkpoints (Paszke et al., 2019; Wightman, 2019) and they are well-tuned separately by their designers. Different from Bai et al. (2021) which uses the same training setting for all models, we follow Paul & Chen (2021) to take different neural network models trained and tuned to the optimum individually rather than using a common but potentially suboptimal setting, since one setting could be biased to some models as well.

Table 1: Comparison of the target models investigated in the main text. More variants (Deit, ViT-SAM, Swin-ViT, etc.) can be found in Appendix B.

| Model | ViT backbone | | | | Pretraining | |
| | Layers | Hidden size | Attention | Params | Pretraining dataset | Scale |
|---|---|---|---|---|---|---|
| ViT-S/16 | 8 | 786 | Self-attention | 49M | ImageNet-21K | 14M |
| ViT-B/16 | 12 | 786 | Self-attention | 87M | ImageNet-21K | 14M |
| ViT-L/16 | 24 | 1024 | Self-attention | 304M | ImageNet-21K | 14M |
| ViT-B/16-Res | 12 | 786 | Self-attention | 87M | ImageNet-21K | 14M |
| T2T-ViT-14 | 14 | 384 | Self-attention | 22M | - | - |
| T2T-ViT-24 | 24 | 512 | Self-attention | 64M | - | - |
| DeiT-S/16 | 12 | 384 | Self-attention | 22M | - | - |
| Dist-DeiT-B/16 | 12 | 768 | Self-attention | 87M | - | - |
| Swin-S/4 | (2,2,18,2) | 96 | Self-attention | 50M | - | - |
| SEResNet50 | - | - | Squeeze-and-Excitation | 28M | - | - |
| ResNeXt-32x4d-ssl | - | - | - | 25M | YFCC100M | 100M |
| ResNet50-swsl | - | - | - | 26M | IG-1B-Targeted | 940M |
| ResNet18 | - | - | - | 12M | - | - |
| ResNet50-32x4d | - | - | - | 25M | - | - |
| ShuffleNet | - | - | - | 2M | - | - |
| MobileNet | - | - | - | 4M | - | - |
| VGG16 | - | - | - | 138M | - | - |

## 3.1 Vision Transformers

For vision transformer, we consider the original ViT (Dosovitskiy et al., 2020) and its variants:

**Vanilla ViT with different training schemes (ViT, DeiT, ViT-SAM)**: The original ViT (Dosovitskiy et al., 2020) mostly follows the original design of Transformer (Vaswani et al., 2017; Devlin et al., 2018) on language tasks. For a 2D image $x_i \in \mathbb{R}^{H \times W \times C}$ ($1 \leq i \leq N$) with resolution $H \times W$ and $C$ channels, it is divided into a sequence of $N = \frac{H \cdot W}{P^2}$ flattened 2D patches of size $P \times P$, $x_i \in \mathbb{R}^{P^2 \cdot C}$ ($1 \leq i \leq N$). The patches are encoded into patch embeddings with a simple convolutional layer, where the kernel size and stride of the convolution is exactly $P \times P$. DeiT (Touvron et al., 2021) further improves the ViT's performance using data augmentation or distillation from CNN teachers with an additional distillation token. We investigate ViT-{S,B,L}/16, DeiT-S/16 and Dist-DeiT-B/16 as defined in the corresponding papers in the main text and discuss other structures in Appendix B. ViT-SAM (Chen et al., 2021) uses sharpness-aware minimization (Foret et al., 2020) to train ViTs from scratch on ImageNet without large-scale pretraining or strong data augmentations. We include ViT-SAM-B/16 in the main text.

**Hybrid of CNN and ViT (CNN-ViT):** Dosovitskiy et al. (2020) also proposed a hybrid architecture for ViTs by replacing raw image patches with patches extracted from a CNN feature map. This is equivalent to adding learned CNN blocks to the head of ViT. We investigate ViT-B/16-Res in our experiments, where the input sequence is obtained by flattening the spatial dimensions of the feature maps from ResNet50.

**Hybrid of T2T and ViT (T2T-ViT):** Yuan et al. (2021) proposed to overcome the limitations of the simple tokenization in ViTs, by progressively structurizing an image to tokens with a token-to-token (T2T) module, which recursively

aggregates neighboring tokens into one token such that low-level structures can be better learned. T2T-ViT was shown to perform better than ViT when trained from scratch on a midsize dataset. We investigate T2T-ViT-14 and T2T-ViT-24 in our experiments.

**Hybrid of shifted windows and ViT (Swin-T):** Liu et al. (2021) computes the representations with shifted windows, which brings greater efficiency by limiting self-attention computation to non-overlapping local windows while also allowing for cross-window connection. We investigate Swin-S/4 in the main text and discuss other structures in Appendix B.

## 3.2 Baselines

We study several CNN models for comparison, including ResNet18 (He et al., 2016), ResNet50-32x4d (He et al., 2016), ShuffleNet (Zhang et al., 2018), MobileNet (Howard et al., 2017), and VGG16 (Simonyan & Zisserman, 2014). We also consider the SEResNet50 model, which uses the Squeeze-and-Excitation (SE) block (Hu et al., 2018) that applies attention to channel dimensions. The recently proposed MLP-Mixer (Tolstikhin et al., 2021) and ConvNeXt (Liu et al., 2022) are also included for comparison.

The aforementioned MLP and CNNs are all trained on ImageNet from scratch. For a better comparison with pre-trained transformers, we also consider two CNN models pre-trained on larger datasets: ResNeXt-32x4d-ssl (Yalniz et al., 2019) pre-trained on YFCC100M (Thomee et al., 2015), and ResNet50-swsl pre-trained on IG-1B-Targeted (Mahajan et al., 2018) using semi-weakly supervised methods (Yalniz et al., 2019), and then fine-tuned on ImageNet.

## 4 Adversarial Robustness Evaluation Methods

We consider the commonly used $\ell_\infty$-norm bounded adversarial attacks to evaluate the robustness of target models. An $\ell_\infty$ attack is usually formulated as solving a constrained optimization problem:

$$\max_{\mathbf{x}^{adv}} \mathcal{L}\left(\mathbf{x}^{adv}, y\right) \quad \text{s.t.} \ \left\|\mathbf{x}^{adv} - \mathbf{x}_0\right\|_\infty \le \epsilon, \tag{1}$$

where $\mathbf{x}_0$ is a clean example with label $y$, and we aim to find an adversarial example $\mathbf{x}^{adv}$ within an $\ell_\infty$ ball with radius $\epsilon$ centered at $\mathbf{x}_0$, such that the loss of the classifier $\mathcal{L}\left(\mathbf{x}^{adv}, y\right)$ is maximized. We consider untargeted attack in this paper, so an attack is successful if the perturbation successfully changes the model's prediction. The attacks as well as a randomized smoothing method used in this paper are listed below.

**Projected Gradient Descent Attack (Madry et al., 2017)** Projected Gradient Decent (PGD) attack (Madry et al., 2017) solves Eq. 1 by iteratively taking gradient ascent:

$$\mathbf{x}_{t+1}^{adv} = Clip_{\mathbf{x}_0, \epsilon}(\mathbf{x}_t^{adv} + \alpha \cdot \text{sgn}\left(\nabla_{\mathbf{x}} J\left(\mathbf{x}_t^{adv}, y\right)\right)), \tag{2}$$

where $\mathbf{x}_t^{adv}$ stands for the solution after $t$ iterations, and $Clip_{\mathbf{x}_0, \epsilon}(\cdot)$ denotes clipping the values to make each $\mathbf{x}_{t+1, i}^{adv}$ within $[\mathbf{x}_{0,i} - \epsilon, \mathbf{x}_{0,i} + \epsilon]$, according to the $\ell_\infty$ threat model. As a special case, Fast Gradient Sign Method (FGSM) (Goodfellow et al., 2014) uses a single iteration with $t = 1$.

**AutoAttack (Croce & Hein, 2020a)** AutoAttack (Croce & Hein, 2020a) is a parameter-free ensemble of diverse attacks, including two variants of PGD attacks (APGD-CE, APGD-DLR), an optimization based attack (FAB(Croce & Hein, 2020b)) and a query based black-box attack (Square Attack(Croce et al., 2019)).

**Transfer Attack** We consider the transfer attack which studies whether an adversarial perturbation generated by attacking the *source* model can successfully fool the *target* model. This test not only evaluates the robustness of models under the black-box setting, but also becomes a sanity check for detecting the obfuscated gradient phenomenon (Athalye et al., 2018). Previous works have demonstrated that single-step attacks like FGSM enjoys better transferability than multi-step attacks (Kurakin et al., 2017). We thus use FGSM for transfer attack in our experiments.

**Frequency-Filtered Attack for Frequency Analysis** In our frequency study, we conduct PGD attack with an additional frequency filter to force the adversarial perturbations being within a specific frequency domain (a high-frequency

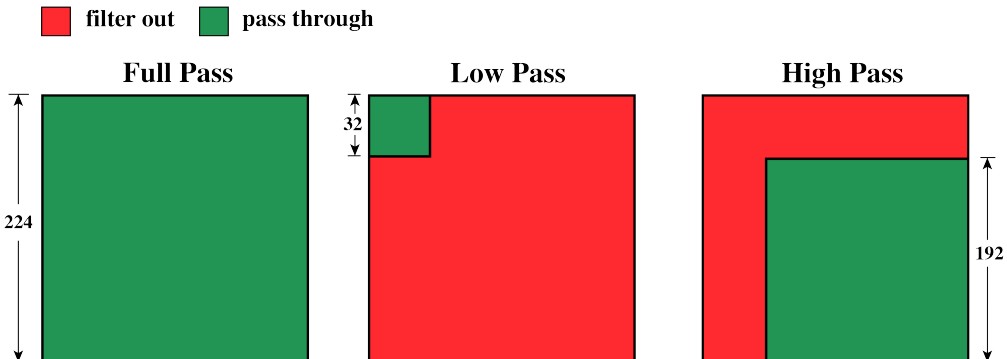

Figure 2: Filters for the frequency-based attack. The frequencies corresponding to the red part are filtered out, and the frequencies corresponding to the green part can pass through. "Full Pass" means all of the frequencies are preserved. "Low Pass" means only low-frequency components are preserved. "High Pass" preserves the high-frequency part.

domain or a low-frequency domain):

$$\mathbf{x}^{adv}_{freq} = \text{IDCT}(\text{DCT}(\mathbf{x}^{adv}_{pgd} - \mathbf{x}_0) \odot \boldsymbol{M}_f) + \mathbf{x}_0, \tag{3}$$

where DCT and IDCT stand for discrete cosine transform and inverse discrete cosine transform respectively, $\mathbf{x}^{adv}_{pgd}$ stands for the adversarial example generated by PGD, and $\boldsymbol{M}_f$ stands for the mask metric defined the frequency filter as illustrated in Figure 2.

**Denoised Randomized Smoothing(Salman et al., 2020) for Certified Robustness Analysis**  We also provide provable certified robustness analysis using denoised randomized smoothing (Salman et al., 2020). The model is certified to be robust with high probability for perturbations within the radius, so the robustness is evaluated as the certified radius. We follow Salman et al. (2020) to train a DnCNN (Zhang et al., 2017) denoiser $\mathcal{D}_\theta$ for each pre-trained model $f$ with stability objective $\mathcal{L}_{Stab}$:

$$\mathcal{L}_{Stab} = \mathbb{E}_{(x_i,y_i)\in\mathcal{D},\delta}\mathcal{L}_{CE}(f(\mathcal{D}_\theta(x_i + \delta)), f(x_i)) \tag{4}$$

where $\delta \sim \mathcal{N}(0, \sigma^2 I)$ follows Gaussian distribution. Then randomized smoothing is applied on the denoised classifier $f \circ \mathcal{D}_\theta$ for robustness certification:

$$g(x) = \arg\max_{c\in\mathcal{Y}} \mathbb{P}[f(\mathcal{D}_\theta(x + \delta)) = c] \;\; \text{where } \delta \sim \mathcal{N}(0, \sigma^2 I). \tag{5}$$

The certified radius (Cohen et al., 2019) is then calculated for the smoothed classifier as:

$$R = \frac{\sigma}{2}(\Phi^{-1}(p_A) - \Phi^{-1}(p_B)), \tag{6}$$

where $\Phi^{-1}$ is the inverse of the standard Gaussian CDF, $p_A = \mathbb{P}(f(x + \delta) = c_A)$ is the confidence of the top-1 predicted class $c_A$, and $p_B = \max_{c\neq c_A} \mathbb{P}(f(x+\delta) = c)$ is the confidence for the second top class. Accordingly, given a perturbation radius, the certified accuracy under this perturbation radius can be evaluated by comparing the given radius to the certified radius $R$.

## 5  Experiments

In this section, we compare the adversarial robustness of ViTs and CNNs from different perspectives: First, we show ViTs are more robust to high-frequency perturbations than CNNs using crafted frequency-filtered attacks and feature visualization.  Second, we conduct a comprehensive empirical study on the comparison between ViTs and CNNs regarding the adversarial robustness against diverse white-box attacks and black-box attacks, and the transferability of the adversarial examples from ViTs to CNNs and vice versa.  Finally, we provide a provable certified robustness comparison using denoised randomized smoothing.

Figure 3: Feature visualization: The learned low-level structure features are highlighted in blue (obviously perceptible) and green (minorly perceptible). The CNNs in the first row learn more low-level features compared with the ViTs in the second row. The ViTs pay more attention to the low-level structures and their feature maps become noisier when ResNet features are introduced (ViT-B/16-Res) or neighboring tokens are aggregated into one token recursively (T2T-ViT-24).

Unless otherwise specified, Clean Accuracy (CA) stands for the accuracy evaluated on the entire ImageNet-1k (Deng et al., 2009) test set, Robust Accuracy (RA) stands for the accuracy on the adversarial examples generated with 1,000 test samples. Higher RA stands for better adversarial robustness. Adversarial training and evaluation results on CIFAR-10 are reported in Appendix F and Appendix C.

## 5.1 Frequency Study Using Filtered-PGD and Feature Visualization

Wang et al. (2020b) has shown that CNN models could take up high-frequency patterns that are almost imperceptible to humans but have distribution correlation with labels, leading to a trade-off between robustness and accuracy. However, we show the original ViT models possess superior adversarial robustness against high-frequency perturbations compared with CNNs. We also show that some variants that introduce non-transformer modules (e.g., ResNet blocks and T2T blocks) to ViTs to improve clean accuracy, can diminish ViTs' original adversarial robustness against high-frequency perturbations, leading to inferior adversarial robustness of hybrid ViTs (e.g., ResViTs and T2T-ViTs). We craft a frequency-filtered PGD attack to study models' resistance to different frequency components. Besides, we use feature visualization to facilitate the illustration of our hypothesis, showing that ViTs are paying less attention to the high-frequency patterns in the images.

Table 2: RA (%) of the target models against frequency-filtered PGD attack in our frequency study. In the "Low-pass" column, only low-frequency adversarial perturbations are preserved and added to the input images. In the "High-pass" column, only high-frequency perturbations are preserved. The "Full-pass" mode preserves all frequency and is the same as the traditional PGD attack. We set the attack step fixed to 40 and vary the attack radius to different values.

| | Low-pass | | | | | High-pass | | | | | Full-pass | | | | |
|---|---|---|---|---|---|---|---|---|---|---|---|---|---|---|---|
| **Attack Radius** | **0.001** | **0.003** | **0.005** | **0.01** | **0.1** | **0.001** | **0.003** | **0.005** | **0.01** | **0.1** | **0.001** | **0.003** | **0.005** | **0.01** | **0.1** |
| ViT-S/16 | 74.0 | 68.1 | 64.7 | 59.8 | 56.2 | 70.8 | 60.7 | 50.6 | 40.4 | 23.4 | 55.4 | 24.6 | 10.2 | 1.0 | 0.0 |
| ViT-B/16 | 71.9 | 64.3 | 60.3 | 55.8 | 49.6 | 66.3 | 53.1 | 44.0 | 33.4 | 21.9 | 48.9 | 14.6 | 6.0 | 0.9 | 0.0 |
| ViT-L/16 | 74.9 | 64.1 | 58.3 | 50.2 | 42.0 | 72.9 | 62.3 | 56.6 | 47.5 | 28.9 | 55.1 | 23.4 | 9.9 | 1.8 | 0.0 |
| ViT-B/16-Res | 83.1 | 81.4 | 80.4 | 79.0 | 75.1 | 62.9 | 29.2 | 16.0 | 7.3 | 3.3 | 45.5 | 8.4 | 2.3 | 0.1 | 0.0 |
| T2T-ViT-14 | 78.0 | 77.2 | 76.0 | 75.8 | 74.3 | 49.6 | 20.5 | 9.1 | 3.1 | 1.4 | 37.1 | 7.0 | 1.8 | 0.0 | 0.0 |
| T2T-ViT-24 | 80.2 | 79.2 | 78.4 | 77.7 | 74.4 | 58.3 | 31.1 | 17.7 | 8.2 | 3.1 | 47.7 | 12.3 | 3.4 | 0.2 | 0.0 |
| MLP-Mixer-B/16 | 69.4 | 64.7 | 62.4 | 60.0 | 59.8 | 56.6 | 32.5 | 19.5 | 6.7 | 1.3 | 34.5 | 3.8 | 0.0 | 0.0 | 0.0 |
| ConvNeXt-S | 80.3 | 79.0 | 77.9 | 76.4 | 74.3 | 53.1 | 24.4 | 14.4 | 6.8 | 6.2 | 15.9 | 0.0 | 0.0 | 0.0 | 0.0 |
| ResNet50-swsl | 78.2 | 74.9 | 73.7 | 71.6 | 72.5 | 45.3 | 12.4 | 5.0 | 2.2 | 3.5 | 24.7 | 2.9 | 1.4 | 0.4 | 0.0 |
| ResNet50-32x4d | 75.0 | 66.3 | 62.7 | 59.0 | 61.5 | 47.7 | 17.1 | 7.4 | 3.3 | 3.5 | 28.2 | 3.2 | 1.2 | 0.4 | 0.1 |

### 5.1.1 Frequency Study

**ViTs are more resistant to high-frequency perturbations and have less bias towards high-frequency features that have spurious correlation with labels.** We design a frequency study to verify our hypothesis that ViTs are adversarially more robust compared with CNNs and MLP-Mixer because ViTs learn less high-frequency features. As defined in equation 3, for adversarial perturbations generated by PGD attack, we first project them to the frequency domain by DCT. We design three frequency filters illustrated in Figure 2: the full-pass filter, the low-pass filter, and the high-pass filter. We take $32 \times 32$ pixels in the low-frequency area out of $224 \times 224$ pixels as the low-pass filter, and $192 \times 192$ pixels in the high-frequency area as the high-pass filter. Each filter allows only the corresponding frequencies to pass through – when the adversarial perturbations go through the low-pass filter, the high-frequency components are filtered out and vice versa, and the full-pass filter makes no change. We then apply these filters to the frequencies of the perturbations, and project them back to the spacial domain with the IDCT. We test the RA under different frequency areas, and show the results in Table 2.

The RA of ViTs are much higher in the "High-pass" column when only the high-frequencies of the perturbations are preserved. In contrast, CNNs show significantly lower RA in the "High-pass" column than in the "Low-pass" column. It reflects that CNNs tend to be more sensitive to high-frequency adversarial perturbations compared to ViTs. As shown in the Table 2, CNNs have moderate robust accuracy drop w.r.t. low-frequency perturbations but severe decrease against high-frequency perturbations when increasing the perturbation rates. For example, when increasing the attack radius from 0.001 to 0.01, the robust accuracy of ResNet50-32x4d decreases from 75.0% to 59.0% (by 16.0%) for low-pass adversarial attack, but 47.0% to 3.3% (by 43.7%) for high-pass adversarial attack, indicating the high-frequency perturbations are the main cause to the low robust accuracy for CNNs. However, when increasing the attack radius from 0.001 to 0.01, ViT-B/16 decreases from 71.9% to 55.8% (by 16.1%) for low-pass adversarial attack, but only 66.3% to 33.4% (by 32.9%) for high-frequency-perturbations, showing much better resistance to high-frequency perturbations compared with CNNs.

**Introducing CNN or T2T blocks makes ViTs less robust to high-frequency perturbations.** One interesting and perhaps surprising finding is that ViTs have worse robustness when modules that claimed to help learning local structures are added ahead of the transformer blocks. For example, T2T-ViT adds several T2T modules to the head of ViT which iteratively aggregates the neighboring tokens into one token in each local perceptive field. ViT-B/16-Res takes the features generated by ResNet as inputs, which has the same effect as incorporating a trained CNN layer in front of the transformer blocks. Both modules help to learn local structures like edges and lines (Yuan et al., 2021). We observe that ResNet and T2T modules that could help improve the CA of the hybrid ViTs makes the models more sensitive to high-frequency perturbations. T2T-ViT-14, T2T-ViT-24 and ViT-B/16-Res have lower RA in the "High-pass" column and higher RA in the "Low-pass" column compared with vanilla ViTs, which correlates with the previous observations that high-frequency features are less adversarially robust. When adding more transformer blocks to the T2T-ViT model, the model becomes less sensitive to the high frequencies of the adversarial perturbations, e.g., the T2T-ViT-24 has an $8.7\%$ higher RA than that of the T2T-ViT-14 in the "High-pass" column.

One possible explanation is that the introduced modules improve the classification accuracy by remembering the high-frequency patterns that repeatedly appear in the training dataset. These structures such as edges and lines are high-frequency and sensitive to perturbations (Wang et al., 2020b). Learning such features makes the model more vulnerable to adversarial attacks. Examination of this hypothesis is conducted though feature visualization.

**Increasing the Proportion of Transformer Blocks Can Improve Robustness** Hendrycks et al. (2019) mentioned that larger model does not necessarily imply better robustness. It can be confirmed by our experiments where ViT-S/16 shows better robustness than larger ViT-B/16 under both PGD attack and AutoAttack. In this case, simply adding transformer blocks to the classifier cannot guarantee better robustness. However, we recognize that for mixed architecture that has both T2T and transformer blocks, it is useful to improve adversarial robustness by increasing the proportion of the transformer blocks in the model. As shown in Table 2 (we will also show similar observations in Tables 3 and 4), T2T-ViT-24 has higher RA than T2T-ViT-14 under both attacks. Besides the transformer block, we find that other attention mechanism modules such as SE block also improves adversarial robustness – as SEResNet50 has the least proportion of attention, the RA of SEResNet50 is lower than ViT and T2T-ViT models but higher than other pure CNNs. These two findings are coherent since the attention mechanism is fundamental in transformer blocks.

### 5.1.2 Feature Visualization

We hypothesis that ViTs are more resistant to high-frequency perturbations because they are learning less high-frequency features that have high correlation with labels but less semantic meanings. To facilitate the illustration, we follow the work of Yuan et al. (2021) to visualize the learned features from the first blocks of the target models in Figure 3. We resize the input images to a resolution of $224 \times 224$ for CNNs and a resolution of $1792 \times 1792$ for ViTs and T2T-ViTs such that the feature maps from the first block are in the same shape of $112 \times 112$. Features with high-frequency patterns like lines and edges are highlighted in blue (obviously perceptible) and green (minorly perceptible). As shown in Figure 3, CNNs like ResNet50-swsl and ResNet50-32x4d learn features with obvious edges and lines. While it is hard to observe such information in the feature maps learned by ViT-B/16.

The frequency study combined with the feature visualization shows that a model's vulnerability against adversarial perturbations is relative to the model's tendency to learn high-frequency low-level features that could have correlation with labels but little semantic information. Techniques that help the model learn such features may improve the accuracy on clean data but at the risk of sacrificing adversarial robustness. We are still facing the trade-off between CA and RA.

## 5.2 Empirical Study: Adversarial Robustness under Various Adversarial Attacks

In this section, we test adversarial robustness against PGD, AutoAttack and transfer attack, and provide observations we drew from this empirical study.

### 5.2.1 Robustness under PGD and AutoAttack

**Settings**   We take attack radius $\epsilon \in \{0.001, 0.003, 0.005, 0.01\}$ for PGD and AutoAttack evaluation. For PGD attack, we fix the attack steps to $n_{iter} = 40$ with other parameters following the default setting of the implementation in Foolbox (Rauber et al., 2020). No hyper-parameter tuning is needed in AutoAttack.

**Results**   We present the results of PGD and AutoAttack in Table 3 and Table 4 respectively. The RA is approximately 0.0% on all the models when $\epsilon = 0.01$ is large, indicating models trained without any adversarial augmentations are vulnerable to large perturbations. However, such vulnerability doesn't hold equally for all models within mild perturbations: For smaller attack radii, **ViT models have higher RA than CNNs under both PGD attack and AutoAttack.** For example, when $\epsilon = 0.001$, the RA for ViT-S/16 is $55.4\%$ while the RA for CNNs is at most $30.0\%$. Moreover, under the same attack radius, the RA of AutoAttack for ViT-S/16 is $48.1\%$ compared to $6.1\%$ of ShuffleNet, which is a large gap. We visualize the clean/robust accuracy tradeoff and model size of these models in Figure 1, and ViT models are noticeably on the upper right over CNN models.

**Introducing ResNet or T2T blocks decreases the RA under both PGD and AutoAttack.** When the features learned by ResNet are introduced, the RA of ViT-B/16 decreases from $48.9\%$ to $45.5\%$ of ViT-B/16-Res under PGD attack, and from $39.8\%$ to $27.7\%$ under AutoAttack, with attack radius $\epsilon = 0.001$. A similar phenomenon can be observed by comparing the RA of ViTs and T2T-ViTs. The RA of T2T-ViT-14 is $18.3\%$ lower under PGD attack and $35.2\%$ lower under AutoAttack compared with the RA of ViT-S/16, under attack radius $\epsilon = 0.001$.

We also verify that pre-training with clean images does not improve robustness. (Dosovitskiy et al., 2020) points out that pre-training is critical for ViTs to achieve competitive standard accuracy with CNNs trained from scratch. However, we note that pre-training doesn't bring better robustness to ViTs. To illustrate this point, we include CNNs pre-trained on large datasets and fine-tuned on ImageNet-1k to check the effect of pre-training on adversarial robustness. CNNs pre-trained on large datasets IG-1B-Targeted (Mahajan et al., 2018) and YFCC100M (Thomee et al., 2015) that are even larger than ImageNet-21k used by ViT, ResNet50-swsl and ResNeXt-32x4d-ssl, still have similar or even lower RA than ResNet18 and ResNet50-32x4d that are not pre-trained. This supports our observation that pre-training in its current form may not be able to improve adversarial robustness, which is also in accordance with Hendrycks et al. (2019). Besides, the results show that SAM can further improve model's adversarial robustness. This is because the SAM objective is formulated as a min-max optimization similar to adversarial training: but instead of adding adversarial perturbations to the input space, SAM adds adversarial perturbations to the weights.

Table 3: Clean Accuracy (%) and Robust Accuracy (%) of target models against 40-step PGD attack with different radii. More results of the SOTA ViTs can be found in Appendix B.

| Attack Radius | CA | RA against PGD | | | |
|---|---|---|---|---|---|
| | | 0.001 | 0.003 | 0.005 | 0.01 |
| ViT-S/16 | 77.6 | **55.4** | **24.6** | **10.2** | 1.0 |
| ViT-B/16 | 75.7 | 48.9 | 14.6 | 6.0 | 0.9 |
| ViT-L/16 | 79.2 | 55.1 | 23.4 | 9.9 | **1.8** |
| ViT-SAM-B/16 | 76.7 | **63.4** | **37.0** | **20.1** | **3.8** |
| ViT-B/16-Res | 84.0 | 45.5 | 8.4 | 2.3 | 0.1 |
| T2T-ViT-14 | 80.1 | 37.1 | 7.0 | 1.8 | 0.0 |
| T2T-ViT-24 | 82.2 | 47.7 | 12.3 | 3.4 | 0.2 |
| Deit-S/16 | 77.7 | 48.9 | 17.6 | 7.1 | 1.1 |
| Dist-Deit-B/16 | 81.8 | 55.6 | 17.7 | 4.5 | 0.4 |
| Swin-S/4 | 81.8 | 40.0 | 12.4 | 3.2 | 0.2 |
| MLP-Mixer-B/16 | 73.8 | 41.9 | 10.7 | 4.3 | 0.4 |
| ConvNeXt-S | 82.7 | 42.4 | 8.1 | 2.6 | 0.0 |
| SEResNet50 | 75.7 | 35.4 | 4.9 | 0.8 | 0.1 |
| ResNeXt-32x4d-ssl | 80.3 | 23.0 | 2.9 | 1.2 | 0.5 |
| ResNet50-swsl | 81.2 | 24.7 | 2.9 | 1.4 | 0.4 |
| ResNet18 | 70.0 | 24.9 | 2.0 | 0.6 | 0.1 |
| ResNet50-32x4d | 77.6 | 28.2 | 3.2 | 1.2 | 0.4 |
| ShuffleNet | 69.4 | 15.0 | 0.6 | 0.2 | 0.0 |
| MobileNet | 71.9 | 16.7 | 0.4 | 0.0 | 0.0 |
| VGG16 | 71.6 | 26.3 | 3.2 | 1.3 | 0.0 |

Table 4: Clean Accuracy (%) and Robust Accuracy (%) of target models against AutoAttack with different attack radii. More results of the SOTA ViTs can be found in Appendix B.

| Attack Radius | CA | RA against AutoAttack | | | |
|---|---|---|---|---|---|
| | | 0.001 | 0.003 | 0.005 | 0.01 |
| ViT-S/16 | 77.6 | **48.1** | 6.0 | 0.5 | 0.0 |
| ViT-B/16 | 75.7 | 39.8 | 5.4 | 0.6 | 0.0 |
| ViT-L/16 | 79.2 | 46.6 | **8.5** | **1.0** | 0.0 |
| ViT-SAM-B/16 | 76.7 | **59.8** | **26.0** | **8.4** | **0.1** |
| ViT-B/16-Res | 84.0 | 27.7 | 0.9 | 0.0 | 0.0 |
| T2T-ViT-14 | 80.1 | 12.9 | 0.1 | 0.0 | 0.0 |
| T2T-ViT-24 | 82.2 | 20.8 | 0.3 | 0.0 | 0.0 |
| Dist-Deit-S/16 | 79.3 | 43.1 | 3.7 | 0.2 | 0.0 |
| Dist-Deit-B/16 | 81.8 | 42.7 | 3.4 | 0.2 | 0.0 |
| Swin-S/4 | 81.8 | 7.9 | 0.1 | 0.0 | 0.0 |
| MLP-Mixer-B/16 | 73.8 | 34.5 | 3.8 | 0.0 | 0.0 |
| ConvNeXt-S | 82.7 | 15.9 | 0.0 | 0.0 | 0.0 |
| SEResNet50 | 75.7 | 21.6 | 0.6 | 0.0 | 0.0 |
| ResNeXt-32x4d-ssl | 80.3 | 6.5 | 0.0 | 0.0 | 0.0 |
| ResNet50-swsl | 81.2 | 8.1 | 0.0 | 0.0 | 0.0 |
| ResNet18 | 70.0 | 14.3 | 0.4 | 0.0 | 0.0 |
| ResNet50-32x4d | 77.6 | 13.2 | 0.2 | 0.0 | 0.0 |
| ShuffleNet | 69.4 | 6.1 | 0.0 | 0.0 | 0.0 |
| MobileNet | 71.9 | 7.8 | 0.0 | 0.0 | 0.0 |
| VGG16 | 71.6 | 16.7 | 0.5 | 0.0 | 0.0 |

### 5.2.2 Transferability of Adversarial Examples from ViTs to CNNs and Vice Versa

We also conduct transfer attack to test the adversarial robustness in the black-box setting as described in Section 4. We consider attacks with $\ell_\infty$-norm perturbation no larger than 0.1 and present the results in Figure 4. When the

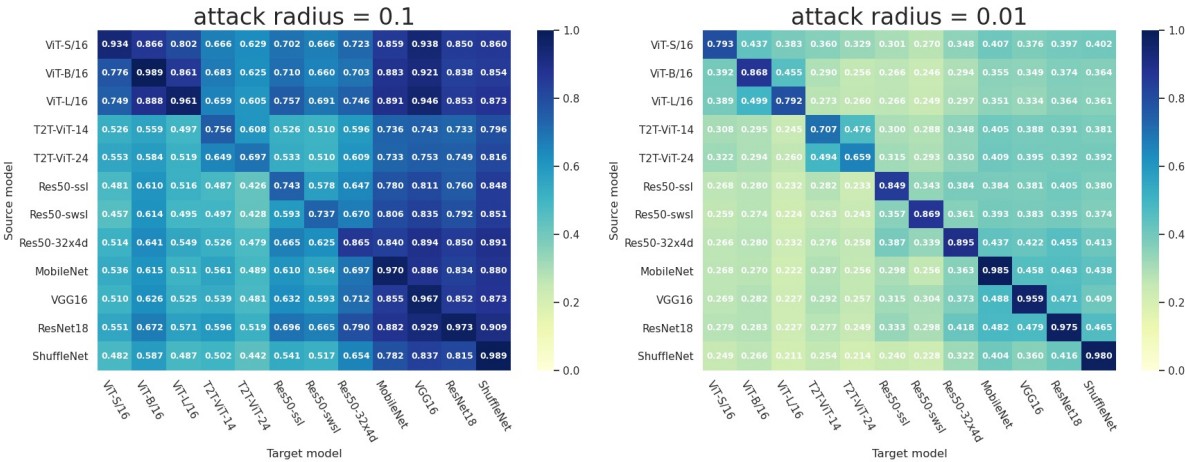

Figure 4: Target model error rate on adversarial examples (i.e., 1.0 - RA) against transfer attack using FGSM with different attack radii. The rows stand for the surrogate models used to generated adversarial examples. The columns stand for the target models. Darker rows correlate to the source models that generate more transferable adversarial examples. Darker columns mean that the target models are more vulnerable to transfer attack."Res50-ssl" and "Res50-swsl" are in short of "ResNeXt-32x4d-ssl" and "ResNet50-swsl" respectively. Results for more radii can be found in Appendix E.

ViTs serve as the target models and CNNs serve as the source models, as shown in the lower left of each subplot, the RER of the transfer attack is quite low. On the other hand, when the ViTs are the source models, the adversarial examples they generate have higher RER when transferred to other target models. As a result, the first three rows and the last seven columns are darker than the others. Besides, for the diagonal lines in the figure where FGSM actually attacks the models in a white-box setting, we can observe that ViTs are less sensitive to attack with smaller radii compared to CNNs, and T2T modules make ViTs more robust to such one-step attack. In addition, adversarial examples transfer well between models with similar structures. As ViT-S/16, ViT-B/16 and ViT-L/16 have similar structures, the adversarial examples generated by them can transfer well to each other, and it is similar for T2T-ViTs and CNNs respectively.

The finding that ViTs rely less on high-frequency patterns also helps us understand why ViT adversarial examples are better transferred to CNNs as shown in Figure 4. As studied in (Ilyas et al., 2019), they found and called those features less prone to be attacked by adversarial attacks as "robust features", which is consistent with our understanding of the "high-level" features that are preferable for neural networks to learn. And those noise-like "non-robust" features are similar to our high-frequency features that contain low-level information. Since the "non-robust" high-frequency patterns are generated with specific models, adversarial perturbations with regard to such patterns should be assumed to be harder to transfer. Since ViTs are less sensitive to such high-frequency patterns, we assume the adversarial perturbations will be forced to rely less on model-specific feature patterns and thus be easier to transfer. It is interesting to notice that ViTs' adversarial examples could be even stronger than the CNN counterparts with large attack radii, and we think it's because the transferable adversarial perturbations need larger attack radii which could be further explored.

## 5.3 Provably Certified Robustness Comparison Using Denoised Randomized Smoothing

**Settings**  We train the denoisers using the stability objective for 25 epochs with a noise level of $\sigma = 0.25$, learning rate of $10^{-5}$ and a batch size of 64. We iterate through the ImageNet dataset to calculate the corresponding radii according to Eq. 6, and report the certified accuracy versus different radii as defined in (Salman et al., 2020) in Table 5.

**Results**  As shown in Table 5 ViT-S/16 has higher certified accuracy than ResNet18 and ResNet50, showing better certified robustness of vision transformers over CNNs. We also found that, in the same settings, training a Gaussian denoiser for the ResNet is harder than for the ViT-S/16. Accuracy of ViT-S/16 with denoiser at noise of $\sigma = 0.25$ is

Table 5: Certified robust accuracy w.r.t. different radii using denoised randomized smoothing (Salman et al., 2020) ($\sigma = 0.25$). Pure ViTs are highlighted with gray shadows.

| Radius | 0.1 | 0.2 | 0.3 | 0.4 | 0.5 | 0.6 | 0.7 | 0.8 | 0.9 |
|---|---|---|---|---|---|---|---|---|---|
| ViT-S/16 | 0.5944 | 0.5452 | 0.4936 | 0.4424 | 0.3972 | 0.3428 | 0.2820 | 0.2044 | 0.0 |
| DeiT-S/16 | 0.6352 | 0.5880 | 0.5380 | 0.4948 | 0.4476 | 0.4004 | 0.3408 | 0.2604 | 0.0 |
| Dist-DeiT-S/16 | 0.6072 | 0.5600 | 0.5176 | 0.4716 | 0.4172 | 0.3620 | 0.3108 | 0.2360 | 0.0 |
| SAM-ViT | 0.6320 | 0.6040 | 0.5520 | 0.5360 | 0.5040 | 0.4600 | 0.4160 | 0.3560 | 0.0 |
| T2T-ViT-14 | 0.4044 | 0.3816 | 0.3580 | 0.3348 | 0.3044 | 0.2660 | 0.2276 | 0.1816 | 0.0 |
| VGG16 | 0.3772 | 0.3220 | 0.2796 | 0.2372 | 0.1964 | 0.1580 | 0.1176 | 0.0768 | 0.0 |
| ResNet50 | 0.4584 | 0.4096 | 0.3604 | 0.3140 | 0.2676 | 0.2208 | 0.1820 | 0.1268 | 0.0 |
| SEResNet50 | 0.4880 | 0.4440 | 0.3880 | 0.3360 | 0.2920 | 0.2680 | 0.2160 | 0.1760 | 0.0 |
| ConvNext-S | 0.5160 | 0.4760 | 0.4320 | 0.3920 | 0.3480 | 0.2880 | 0.2440 | 0.1880 | 0.0 |

64.84% (4.996% without any denoiser), while accuracy of ResNet50 and ResNet18 with denoiser at the same noise is 47.782% (5.966% without any denoiser).

**Pure ViTs possess better certified robust accuracy than CNNs.** As shown in Table 5, pure ViTs (ViT-S/16, DeiT-S/16, Dist-DeiT-S/16 and SAM-ViT) have higher certified robust accuracy than CNNs. Introducing T2T blocks to ViTs can cause the model to have inferior certified robust accuracy even than CNNs especially for small radii, e.g., radii smaller than 0.5. While sharpness-aware minimization helps further improve ViTs' certified robust accuracy.

**Modern CNN design helps bridge the performance gap between CNNs and ViTs.** The design of modern non-transformer models, e.g. ConvNext, has borrowed many techniques from transformers. For example, using larger kernel size to imitate the global attention mechanism of transformers, following the transformers to change stem to "Patchify", using invertible bottleneck as transformers do, substituting BN with LN, replacing ReLU with GeLU, etc. All these changes are targeted to imitate the transformer's operation without introducing the attention blocks. MLP-Mixer also does similar modifications. Our experiments (Table 3, Table 4 and Table 5) show that such modification helps bridge the performance gap between CNNs and transformers not only in terms of clean accuracy, but also empirical and certified robust accuracy. We also show that CNNs with attention mechanism, i.e. SEResNet50 in our experiments, also has better certified robustness than CNNs.

### 5.4 Extended Analysis

In Appedix A, we also conduct a sanity check to verify that ViT's improvement is not caused by insufficient attack optimization, and an explanation from Hopfield network perspective is provided. Besides, we verify that adversarial training could be directly applied to ViTs in Appendix F.

## 6 Conclusion

This paper presents a comprehensive study on the robustness of ViTs against adversarial perturbations. Our results indicate that ViTs are more robust than CNNs on the considered adversarial attacks and certified robustness settings. We show that the features learned by ViTs contain less low-level information, contributing to improved robustness against adversarial perturbations that often contain high-frequency components. Also, introducing convolutional blocks in ViTs can facilitate learning low-level features but has a negative effect on adversarial robustness and makes the models more sensitive to high-frequency perturbations. Moreover, both the sanity performance and the (certified and empirical) adversarial robustness are improved in the modern CNN designs that leverage techniques from ViTs to imitate the global attention behavior. We also demonstrate adversarial training for ViT. Our work provides a deep understanding of the intrinsic robustness of ViTs and can be used to inform the design of robust vision models based on the transformer structure.

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

## Supplemental Material

In this supplemental material, we provide more analysis and results in our experiments.

## A    The Source of Adversarial Robustness

In this section we examine the source of the adversarial robustness revealed in our experiments.

**The improved robustness of ViT is not caused by insufficient attack optimization.**    We first demonstrate that the better robustness of ViTs in white-box attacks is not caused by the difficult optimization in ViT by plotting the loss landscape with sufficient attack steps.

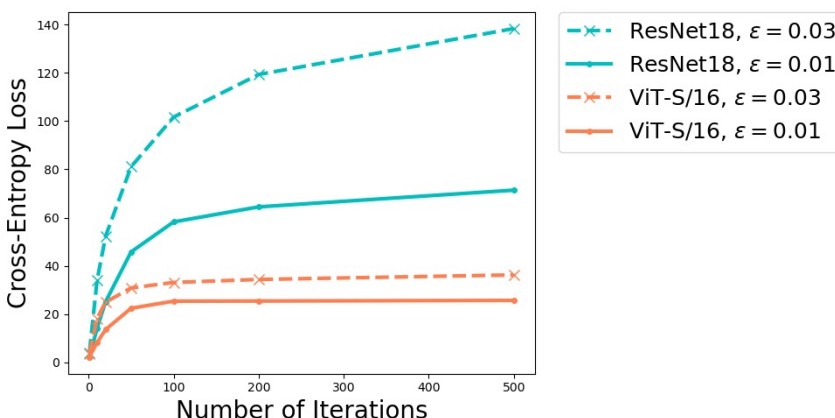

Figure 5: Cross entropy loss versus varying PGD attack steps for ViT-S/16 and RestNet18. The dashed lines corresponds to larger attach radius of 0.03 and the full lines to smaller attack radius of 0.01.

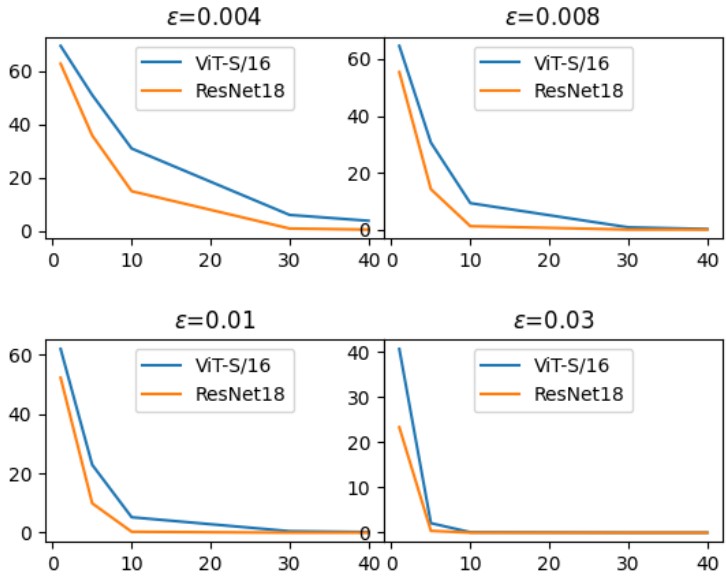

Figure 6: Robust accuracy versus varying PGD attack steps. The attack radii used for evaluation are shown in subtitles.

Figure 5 shows the cross entropy loss versus varing PGD attack steps for ViT-S/16 and ResNet18. Figure 6 shows the robust accuracy versus varing PGD attack steps. As shown in the figures, ViT's loss curves converge at a much lower value than RestNet18, suggesting that the improved robustness of ViT is not caused by insufficient attack optimization.

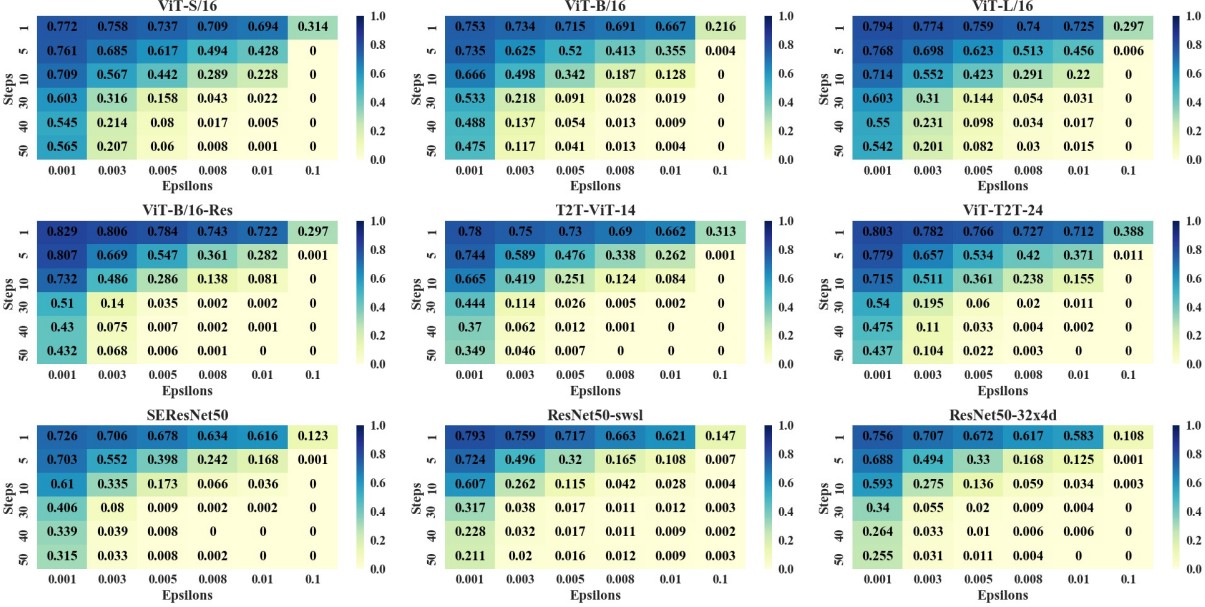

Figure 7: Adversarial accuracy of the target models against PGD attack with different attack radii ("eps") and attack steps ("steps"). When the attack radius and attack steps are increased, the adversarial accuracy of the target model decreases to zero. Darker blocks stand for more robust models against PGD attack.

Figure 7 shows the robust accuracy of more target models against PGD attack with different attack radii ("eps") and attack steps ("steps"). Vision transformers have darker blocks than CNNs', which stands for their superior adversarial robustness against PGD attack.

**A Hopfield Network Perspective** The equivalence between the attention mechanism in transformers to the modern Hopfield network (Krotov & Hopfield, 2016) was recently shown in (Ramsauer et al., 2020). Furthermore, on simple Hopfield network (one layer of attention-like network) and dataset (MNIST), improved adversarial robustness was shown in (Krotov & Hopfield, 2018). Therefore, the connection of attention in transformers to the Hopfield network can be used to explain the improved adversarial robustness for ViTs.

# B Experiments on SOTA ViT Structures

In this section, we supplement the experimental results of recently proposed SOTA ViTs.

**Swin-Trasnformer (Liu et al., 2021)** computes the representations with shifted windows scheme which brings greater efficiency by limiting self-attention computation to non-overlapping local windows while also allowing for cross-window connection.

**DeiT (Touvron et al., 2021)** further improves the ViTs' performance using data augmentation or distillation from CNN teachers with an additional distillation token.

**SAM-ViT (Chen et al., 2021)** uses sharpness-aware minimization (Foret et al., 2020) to train ViTs from scratch on ImageNet without large-scale pretraining or strong data augmentations.

Table 6 summarizes the information of models investigated in our experiments. The window size of the swin transformers in Table 6 is 7. The pre-trained weights of these models are available in `timm` package.

Table 6: SOTA ViT models investigated in our experiments.

| Model | Layers | Hidden size | Heads | Params |
|---|---|---|---|---|
| Deit-T/16 (Touvron et al., 2021) | 12 | 192 | 3 | 6M |
| Deit-S/16 (Touvron et al., 2021) | 12 | 384 | 6 | 22M |
| Deit-B/16 (Touvron et al., 2021) | 12 | 768 | 12 | 87M |
| Dist-Deit-T/16 (Touvron et al., 2021) | 12 | 192 | 3 | 6M |
| Dist-Deit-S/16 (Touvron et al., 2021) | 12 | 384 | 6 | 22M |
| Dist-Deit-B/16 (Touvron et al., 2021) | 12 | 768 | 12 | 87M |
| ViT-SAM-B/16 (Chen et al., 2021) | 12 | 768 | 12 | 87M |
| ViT-SAM-B/32 (Chen et al., 2021) | 12 | 768 | 12 | 88M |
| Swin-T/4 (Liu et al., 2021) | (2,2,6,2) | 96 | (3,6,12,24) | 28M |
| Swin-S/4 (Liu et al., 2021) | (2,2,18,2) | 96 | (3,6,12,24) | 50M |
| Swin-B/4 (Liu et al., 2021) | (2,2,18,2) | 128 | (4,8,16,32) | 88M |
| Swin-L/4 (Liu et al., 2021) | (2,2,18,2) | 192 | (6,12,24,48) | 197M |

Table 7: Robust accuracy (%) of ViTs described in Table 6 against 40-step PGD attack with different attack radii, and also the clean accuracy ("Clean"). A model is considered to be more robust if the robust accuracy is higher.

| Model | Clean | 0.001 | 0.003 | 0.005 | 0.01 |
|---|---|---|---|---|---|
| Deit-T/16 | 72.3 | 36.8 | 8.3 | 2.6 | 0.3 |
| Deit-S/16 | 77.7 | 48.9 | 17.6 | 7.1 | 1.1 |
| Deit-B/16 | 81.3 | 46.6 | 14.3 | 6.0 | 0.9 |
| Dist-Deit-T/16 | 74.4 | 40.6 | 5.7 | 0.7 | 0.2 |
| Dist-Deit-S/16 | 79.3 | 52.4 | 15.1 | 4.3 | 0.3 |
| Dist-Deit-B/16 | 81.8 | 55.6 | 17.7 | 4.5 | 0.4 |
| ViT-SAM-B/16 | 76.7 | 63.4 | 37.0 | 20.1 | 3.8 |
| ViT-SAM-B/32 | 63.8 | 53.2 | 32.3 | 19.7 | 3.1 |
| Swin-T/4 | 78.8 | 33.5 | 6.0 | 1.2 | 0.1 |
| Swin-S/4 | 81.8 | 40.0 | 12.4 | 3.2 | 0.2 |
| Swin-B/4 | 82.3 | 38.8 | 11.1 | 4.1 | 0.3 |
| Swin-L/4 | 84.2 | 38.7 | 11.1 | 2.9 | 0.4 |

Table 7 shows the clean and robust accuracy of ViTs in Table 6 against 40-step PGD attack with different radii. And results for AutoAttack are shown in Table 8. Swin-transformers introduce shifted windows scheme that limit self-attention computation to non-overlapping local windows, which harms the robustness as Tokens-to-Token scheme according to the above results.

## C Experiments on Cifar-10

We choose the ImageNet as the benchmark because ViTs can hardly converge when training directly on small datasets like Cifar. Therefore, we finetune the ViTs instead. As shown in Table 9, ViT-B/4 performs higher robust accuracy than WideResNet, which is consistent with the trend on ImageNet.

Table 8: Robust accuracy (%) of ViTs described in Table 6 against AutoAttack with different attack radii, and also the clean accuracy ("Clean"). A model is considered to be more robust if the robust accuracy is higher.

| Model | Clean | 0.001 | 0.003 | 0.005 | 0.01 |
|---|---|---|---|---|---|
| Deit-T/16 | 72.3 | 23.4 | 0.5 | 0.0 | 0.0 |
| Deit-S/16 | 77.7 | 30.2 | 1.2 | 0.0 | 0.0 |
| Deit-B/16 | 81.3 | 20.4 | 0.3 | 0.1 | 0.0 |
| Dist-Deit-T/16 | 74.4 | 31.1 | 0.8 | 0.1 | 0.0 |
| Dist-Deit-S/16 | 79.3 | 43.1 | 3.7 | 0.2 | 0.0 |
| Dist-Deit-B/16 | 81.8 | 42.7 | 3.4 | 0.2 | 0.0 |
| ViT-SAM-B/16 | 76.7 | 59.8 | 26.0 | 8.4 | 0.1 |
| ViT-SAM-B/32 | 63.8 | 48.9 | 23.6 | 9.7 | 0.8 |
| Swin-T/4 | 78.8 | 6.8 | 0.1 | 0.0 | 0.0 |
| Swin-S/4 | 81.8 | 7.9 | 0.1 | 0.0 | 0.0 |
| Swin-B/4 | 82.3 | 2.4 | 0.1 | 0.0 | 0.0 |
| Swin-L/4 | 84.2 | 4.3 | 0.1 | 0.0 | 0.0 |

Table 9: Robust accuracy of ViT-B/4 and WideResNet against PGD-10 attack with different attack radii.

| Model | 0.001 | 0.003 | 0.01 | 0.03 |
|---|---|---|---|---|
| ViT-B/4 | 0.9202 | 0.6242 | 0.0994 | 0.0103 |
| WideResNet | 0.7744 | 0.5923 | 0.0854 | 0.0000 |

## D  Robustness Against Adversarial Deformation

Besides additively perturbing the correctly classified image, ADef (Alaifari et al., 2018) iteratively applies small deformations to the clean data. We show the robust accuracy against such perturbations in Table 10, which is in accordance to the results of PGD and AutoAttack.

Table 10: Robust accuracy (%) against AFef under the default setting described in Alaifari et al. (2018).

| Model | ViT-S/16 | VGG16 | DenseNet | MobileNet | ResNet18 |
|---|---|---|---|---|---|
| **Robust Accuracy** | **12.4** | 10.8 | 11.1 | 11.7 | 11.8 |

## E  Transfer Attack Results

Transfer attack results using more attack radii are provided in Figure 8

## F  Adversarial Training

**Settings** We also conduct a preliminary experiment on adversarial training for ViT. For this experiment we use CIFAR-10 (Krizhevsky et al., 2009) with $\epsilon = 8/255$ and the ViT-B/16 model. Since originally this ViT was pre-trained on ImageNet with image size $224 \times 224$ and patch size $16 \times 16$ while image size on CIFAR-10 is $32 \times 32$, we downsample the weights for patch embeddings and resize patches to $4 \times 4$, so that there are still $8 \times 8$ patches and we name the new model as ViT-B/4. Though ViT originally enlarged input images on CIFAR-10 for natural fine-tuning and evaluation, we keep the input size as $32 \times 32$ to make the attack radius comparable. For training, we use PGD-7 (PGD with 7 iterations) (Madry et al., 2017) and TRADES (Zhang et al., 2019) methods respectively, with no additional data during adversarial training. We compare ViT with two CNNs, ResNet18 (He et al., 2016) and WideResNet-34-10 (Zagoruyko & Komodakis, 2016). To save training cost, we train each model for 20 epochs only, although some prior works used around hundreds of epochs (Madry et al., 2017; Pang et al., 2020) and are very costly for large models. We use a batch size of 128, an initial learning rate of 0.1, an SGD optimizer with momentum 0.9,

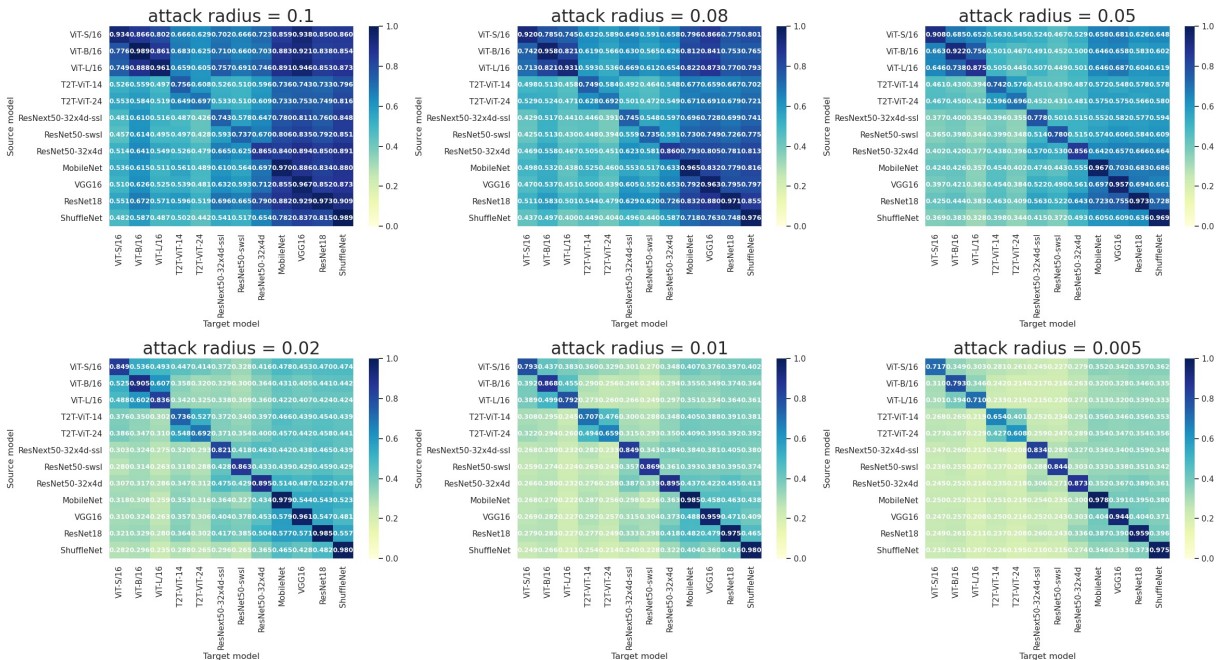

Figure 8: ASR of transfer attack using FGSM with different attack radii. The rows stand for the surrogate models used to generated adversarial examples in the white-box attack approach. The columns stand for the target models. Darker rows correlate to the source models that generate more transferable adversarial examples. While darker columns mean that the target models are more vulnerable to the transfer attack. "Res50-ssl" and "Res50-swsl" are in short of "ResNeXt-32x4d-ssl" and "ResNet50-swsl" respectively.

Table 11: Results of adversarial training for different models using PGD-7 (7-step PGD attack) and TRADES respectively on CIFAR-10. ViT-B/4 is a variant of ViT-B/16 where we downsample the patch embedding kernel from $16 \times 16$ to $4 \times 4$ to accommodate the smaller image size on CIFAR-10. We report the clean accuracy (%) and robust accuracy (%) evaluated with PGD-10 and AutoAttack respectively. Each model is trained using only 20 epochs to reduce the cost.

| Model | Method | Clean | PGD-10 | AutoAttack |
|---|---|---|---|---|
| PreActResNet18 | PGD-7 | 77.3 | 48.9 | 44.4 |
| | TRADES | 77.6 | 49.4 | 44.9 |
| WideResNet-34-10 | PGD-7 | 80.3 | 52.2 | 48.4 |
| | TRADES | 81.6 | 53.4 | 49.3 |
| ViT-B/4 | PGD-7 | 85.9 | 51.7 | 47.6 |
| | TRADES | 85.0 | 53.9 | 49.2 |

and the learning rate decays after 15 epochs and 18 epochs respectively with a rate of 0.1. While we use a weight decay of $5 \times 10^{-4}$ for CNNs as suggested by Pang et al. (2020) that $5 \times 10^{-4}$ is better than $2 \times 10^{-4}$, we still use $2 \times 10^{-4}$ for ViT as we find $5 \times 10^{-4}$ causes an underfitting for ViT. We evaluate the models with PGD-10 (PGD with 10 iterations) and AutoAttack respectively.

**Results** We show the results in Table 11. The ViT model achieves higher robust accuracy compared to ResNet18, and comparable robust accuracy compared to WideResNet-34-10, while ViT achieves much better clean accuracy compared to the other two models. Here ViT does not advance the robust accuracy after adversarial training compared to large CNNs such as WideResNet-34-10. We conjecture that ViT may need larger training data or longer training

epochs to further improve its robust training performance, inspired by the fact that on natural training ViT is not able to perform well either without large-scale pre-training. And although T2T-ViT improved the performance of natural training when trained from scratch, our previous results in Table 3 and Table 4 show that the T2T-ViT structure may be inherently less robust. We have also tried Wong et al. (2020) which was proposed to mitigate the overfitting of FGSM to conduct fast adversarial training with FGSM, but we find that it can still cause catastrophic overfitting for ViT such that the test accuracy on PGD attacks remains almost 0. We conjecture that this fast training method may be not suitable for pre-trained models or require further adjustments. Our experiments in this section demonstrate that the adversarial training framework with PGD or TRADES is applicable for transformers on vision tasks, and we provide baseline results and insights for future exploration and improvement.

