# OpenReview forum: "On the Adversarial Robustness of Vision Transformers"
_TMLR — Accepted by TMLR_

### Review · Reviewer_N2u2 · 2022-06-13

**Summary Of Contributions:**

Summary: This work studies the adversarial robustness of vision transformers. The authors first show that ViTs possess better adversarial robustness than MLP-Mixer, CNNs, and ConvNeXt. Furthermore, the authors provided three empirical observations to explain why ViTs are more robust.

**Broader Impact Concerns:**

I have no particular concerns regarding the ethical implications of this work.

**Requested Changes:**

Remove the trivial Adversarial Training experiments or Make the experiments more comprehensive.
Reorganize the paper to better present the main findings.

**Strengths And Weaknesses:**

Strengths:
1. The topic of this paper is very relevant to the community.
2. The paper is easy to follow.
3. The study from the perspective of frequency is interesting.

Weaknesses:
1. The main claim of this paper is ViTs are more robust than CNNs, Mixer-MLP. The claim is only true when a very small perturbation range is allowed. When a large perturbation is applied, they are all equally vulnerable.

2. The third observation that Adversarial training is also applicable to ViT for training robust models is trivial. The claim that pre-training without adversarial training on larger datasets does not significantly improve adversarial robustness is simply an observation, which cannot explain the core claim that ViTs are more robust.

3. In Table 2, the author report Attack Success Rate (%) on different models, which is inappropriate. The author evaluates the clean accuracy of each model on the whole test set of ImageNet-1k, which includes the false classified images even under no perturbation. Hence, The reported score is not true ASR, which can be misleading.

4. This work studies the robustness of vision transformers. However, some points are supported by weak experiments. For instance, the adversarial training experiments are conducted on CIFAR10. The contribution is not solid given this is an empirical study.

Given the limited novelty and the weakness above. This is not good enough for acceptance.

---

> ### Author Response · Authors · 2022-08-03
> **Response to Reviewer N2u2**
>
> Thanks for the comments. We have uploaded a revised version with the following changes:
>
> **Revised terms**:
>
> * We reorganized the whole experimental sections (Section 4, 5, 6 in the original submission), and merged the Section 5 and Section 6 together such that each analysis is directly following the corresponding results. And we rewrite and highlight the main findings with bold text to make the paper easier to follow. We also put the frequency study in front of the empirical study such that the reasoning about the experimental results could refer to our frequency hypothesis. We also added more models to the certified robustness analysis.
>
> * We revised the “ASR” with “Robust Accuracy” to make it a more appropriate description as one of our evaluation metrics. And we thank the reviewer for pointing this out.
>
> * Following the reviewer’s suggestion, we moved the adversarial training section to the appendix, and rearranged the experiment section, highlighting our important hypothesis and observations. Some side take-aways or verification of some common properties shared by CNNs are only briefly mentioned in the new version or moved to the appendix since they’re not our main contributions.
>
> **Responses to the comments**:
>
> **Q1**: The claim is only true when a very small perturbation range is allowed.
>
> **A1**: Adversarial training technique, i.e., including adversarial perturbations in the training process, is proven to be one of the most effective ways to enhance the adversarial robustness of deep neural networks. However, it usually relies on additional computation and incurs a trade-off between sanity accuracy and robust accuracy. And smaller radii are also commonly used even for adversarial training on ImageNet dataset that is much larger than general dataset like Cifar10. Our empirical study shows ViTs have better intrinsic adversarial robustness than CNNs without any defensive strategy. And we show this could be attributed to the properties of ViTs’ learned features as in the frequency study. Though ViTs are still vulnerable to vigorous perturbations, their superior robustness in the small perturbation range shows the possibility to design better structures to learn intrinsic adversarially robust models. And we also show in the certified robustness analysis that with denoising, the ViTs still possess better certified robustness.
>
> **Q2**: The third observation that Adversarial training is also applicable to ViT for training robust models is trivial. The claim that pre-training without adversarial training on larger datasets does not significantly improve adversarial robustness is simply an observation, which cannot explain the core claim that ViTs are more robust.
>
> **A2**: We understand the reviewer’s concern. An earlier and public version of our work is actually the first study to show that adversarial training also works for ViT models, in addition to CNNs. So we thought it would be helpful to inform the community about our “new” findings. Following your suggestion, we have moved this finding to the appendix. We mentioned the pre-training observation because some previous works have the statement that “pre-training helps improve the adversarial robustness of CNNs”, but their pre-training actually involved adversarial training which can cause confusion. And some ViTs are pre-trained on larger datasets, so we clarify this observation also for fair comparison in the complete results.
>
> **Q3**: In Table 2, the author report Attack Success Rate (%) on different models, which is inappropriate.
>
> **A3**: The appropriate statement should be “Robust Accuracy” instead of ASR. We have revised the term in the new version. This doesn’t affect our claims. Thanks for pointing it out.
>
> **Q4**: The adversarial training experiments are conducted on CIFAR10.
>
> **A4**: The general adversarial training is expensive, especially for the larger scale ViTs. Previous works on adversarial training also mainly focus on small datasets like CIFAR10. Since this is not one of our most intriguing findings, we have moved the corresponding section to the appendix.

---

### Review · Reviewer_UVpe · 2022-06-15

**Summary Of Contributions:**

This work studies the adversarial robustness of vision transformers (ViTs) from various angles. The adversarial robustness is studied through various experiments: white-box attacks, transfer attacks, black-box attacks, as well as certified robustness. The main finding is that “ViTs possess better adversarial robustness when compared with MLP-Mixer and convolutional neural networks”. This finding is further analyzed through a frequency study and a feature visualization. Additionally, the authors showed that adversarial training is applicable to ViTs.

**Broader Impact Concerns:**

No broader impact concerns.

**Requested Changes:**

This work is a collection of robustness experiments on ViTs. However, the experiments fail to tell a coherent story, with many contradicting results. Additionally, due to the diversity of the experiments, they were not performed and evaluated in-depth, leaving many open questions. I hope the authors can address my points in the weakness section and convince me of the unique contributions of their work and how they relate to existing literature.

**Strengths And Weaknesses:**

## Strengths
1. The investigation of adversarial robustness is an important topic, and it is interesting to evaluate novel architectures, such as ViT, MLP-Mixer, ConvNeXt-S.
2. This work provides an extensive collection of robustness experiments on ViTs.
3. As far as I can tell, this work indeed presents the first study that investigates the certified adversarial robustness of ViTs, which is interesting.

## Weaknesses
1. The focus of this work is scattered. For nearly every part of the work, a separate work could be done. Specific examples:
(a) While this might be the first work that evaluates the certified robustness of ViTs, the evaluation is limited to only three models. It would have been great, if more models would have been evaluated, such as ViT-B/16, ViT-L/16, T2T-ViT-14, ConvNeXt-S and SEResNet50.
(b) As the authors state, the adversarial training part is only a preliminary experiment and its results section raises more questions than it provides insights.
In fact, I noticed that for most of the discussed points, dedicated works have been published, which go into much more detail and provide much more insight.

2. A large body of similar works exists, and it is hard to differentiate what makes this work unique from other works. This discussion should not take place in Appendix A, but should be presented in the main manuscript, since it is crucial for the reader to understand, why reading this work is valuable. Especially, the following similarities stood out to me, which should be discussed. It should be clearly stated, how this work distinguishes itself in terms of technical novelty, but also in terms of insight, which cannot be observed in the other works.
(a) The phenomenon that vision transformers are more robust against adversarial perturbations (Figure 1, Table 2) has been widely observed in most other works.
(b) Transfer Attacks for ViTs have been discussed in [Naseer et al. (2021b), A, Mahmood et al. (2021), D].
(c) Adversarial Training of ViTs has been discussed in [B] and many works exist on the general robustness of ViTs [C, Salman et al. (2021)].
(d) A similar frequency study with similar observations has been performed in [Paul & Chen, D].

3. Some results appear to be contradicting:
(a) The abstract states: “ViTs possess better adversarial robustness when compared with MLP-Mixer and convolutional neural networks (CNNs) including ConvNeXt.” However in Table 2: ConvNeXt (a modern convolutional neural network) is more robust than T2T-ViT-14? This contradicts the above claim and I am missing an in-depth discussion regarding this observation (exceeding the one provided in Sec. 6.1).
(b) Similar to (a) and more prominent, in Table 3 the CNNs ConvNeXt-S, SEResNet50, ResNet18, ResNet50-32x4d, and VGG16 appear all to be more robust than the transformer architectures T2T-ViT-14 and Swin-S/4.
(c) While “ViT-S/16 shows better robustness than larger ViT-B/16”, the even larger model Vit-L/16 also shows better robustness than ViT-B/16, and is on par or even more robust than ViT-S/16, which contradicts the explanations in the paper.

4. The authors mention a possible explanation for the contradicting results in Table2 and Table3, where at least ConvNeXt is more robust than T2T-ViT-14 as the “remembering the low-level structures that repeatedly appear in the training dataset”, which is accompanied by visualization in Section 6.1. Unfortunately, this remains only a possible explanation without a definite answer, a proof, or more convincing experiments to support this explanation. Also, experiments to rule out other possibilities are missing.

5. The authors claim that low-level features are less adversarially robust, which is linked to the frequency study. However, the link between frequency, robustness, and low-level features is not clear to me. I hope the authors could provide a proof, an extended discussion with supporting citations, or an extensive supportive experimental evaluation.

[Naseer et al. (2021b)] On Improving Adversarial Transferability of Vision Transformers
[A] Towards Transferable Adversarial Attacks on Vision Transformers; AAAI 2022
[Salman et al. (2021)] Certified patch robustness via smoothed vision transformers
[B] Pyramid Adversarial Training Improves ViT Performance; ArXiv 2021
[C] Towards Robust Vision Transformer; CVPR 2022
[Salman et al. (2021)] Certified patch robustness via smoothed vision transformers
[Paul & Chen (2021)] Vision Transformers are Robust Learners
[Mahmood et al. (2021)] On the Robustness of Vision Transformers to Adversarial Examples
[D] Adversarial Robustness Comparison of Vision Transformer and MLP-Mixer to CNNs; BMVC 2021

---

> ### Author Response · Authors · 2022-08-03
> **Response to Reviewer UVpe (1/N)**
>
> Thanks for the detailed comments. We have uploaded a revised version with the following changes:
>
> **Revised terms**:
>
> * We rearranged the experiment and reasoning sections (the Section 5 and 6 in the original submission, merged into Section 5 in the revised version), making each reasoning statement and key observations directly follow the corresponding experimental results instead of separately. We assume the confusion to the reviewer was mainly caused by the original separate placement of two sections, and the lack of highlight of some explanatory sentences. Therefore, we added highlights and recap terms for these key observations to make our paper easier to follow in the revised version.
>
> * We supplemented more models as suggested in the certified robustness evaluation and have updated the corresponding results in Table 5 of the revised version. From the table, the conclusion that ViT and its variants process better adversarial robustness still holds with provable guarantee.
>
> * We moved the adversarial training section to the appendix and the discussion of concurrent works to the introduction section. We also supplemented two new concurrent works as mentioned in the discussion.
> We added a discussion about the modern CNN designs (MLP-Mixer and ConvNext) in Section 5.3 of our revised version. We note that these modern designs borrowed many design principles from transformers. And we show in our experiments that such improvement not only bridges the gap between CNNs and Transformers in terms of clean accuracy, but also certified and empirical robust accuracy.
>
> **Responses to the comments**:
>
> **Q1**: While this might be the first work that evaluates the certified robustness of ViTs, the evaluation is limited to only three models. It would have been great, if more models would have been evaluated, such as ViT-B/16, ViT-L/16, T2T-ViT-14, ConvNeXt-S and SEResNet50.
>
> **A1**: We added more models for the certified robustness verification. We supplement the new results in Table 5 in the revised version. The key observation holds that ViTs are generally more robust than CNNs in denoised certified robustness verification. As the training of each denoiser on ImageNet takes a lot of time, we tried our best to supplement more models. We also plan to opensource our code so future studies and models can benefit from our findings.
>
> **Q2**: As the authors state, the adversarial training part is only a preliminary experiment and its results section raises more questions than it provides insights.
>
> **A2**: Following the reviewer’s suggestion, we have moved the adversarial training part to the appendix in the updated version.
>
> **Q3**: The discussion of concurrent works should not take place in Appendix A, but should be presented in the main manuscript, since it is crucial for the reader to understand why reading this work is valuable.
>
> **A3**: We moved the discussion of concurrent works to the introduction section in the updated version.
>
> **Q4**: The phenomenon that vision transformers are more robust against adversarial perturbations (Figure 1, Table 2) has been widely observed in most other works. Adversarial Training of ViTs has been discussed in [B] and many works exist on the general robustness of ViTs [C].
>
> **A4**: We have discussed our contributions and concurrent works including [2,3,4,5,6] in the appendix and moved this section to the introduction in the updated version. We would like to note that the references [2-9] actually cited our work (our arxiv version) and acknowledged our contributions and novelty as preceding studies. We thank the reviewer for providing [8,9] which we didn’t cover in the previous discussion and have supplemented a discussion on these two in the updated version. And noticeably, both [8,9] cited our work and we believe our work has provided useful insight for them.
>
> **Q5**:  Transfer Attacks for ViTs have been discussed in [Naseer et al. (2021b), A, Mahmood et al. (2021), D]. A similar frequency study with similar observations has been performed in [Paul & Chen, D].
>
> **A5**: Although our arxiv version and codes were released one and a half months earlier prior to the submission deadline of their workshop version and 6 months earlier prior to the first arxiv version of [7, D], based on our communication with the authors, they mentioned our and their works are concurrent studies and actually cited our work in the publication. Therefore, we don’t think our contribution should be diminished or underrated in this regard. If the reviewer deems necessary, we are happy to share our communication with the authors via the action editor.

---

> ### Author Response · Authors · 2022-08-03
> **Response to Reviewer UVpe (2/N, N=2)**
>
> **Q6**: The abstract states: “ViTs possess better adversarial robustness when compared with MLP-Mixer and convolutional neural networks (CNNs) including ConvNeXt.” However in Table 2: ConvNeXt (a modern convolutional neural network) is more robust than T2T-ViT-14? This contradicts the above claim and I am missing an in-depth discussion regarding this observation.
>
> **A6**: In Section 6 of the original submission (moved to Section 5.1.1 of the revised version), we discussed the effect of introducing T2T modules to the ViT structures which helps training ViTs from scratch but at the same time sacrifices the adversarial robustness of the vanilla ViTs. This effect can also be observed on Res-ViT where CNN modules are introduced. We found in our frequency study and feature visualization that both CNN and T2T modules learn and rely more on high-frequency features which are more vulnerable to adversarial perturbations, making the T2T-ViT and ResViT less robust. Besides, the design of modern non-transformer models, e.g. ConvNext, has borrowed many techniques from transformers. For example, using larger kernel size to imitate the global attention mechanism of transformers, following the transformers to change stem to “Patchify”, using invertible bottleneck as transformers do, substituting BN with LN, fewer normalization layers, fewer activation functions, replacing ReLU with GeLU, etc. All these changes are targeted to imitate the transformer’s operation without introducing the attention blocks. MLP-Mixer also did similar modifications. Our experiments show that such modification does bridge the performance between CNNs and transformers not only in terms of clean accuracy, but also (certified) robust accuracy. We have also included this discussion in Section 5.3 of our revised version.
>
> **Q7**: While “ViT-S/16 shows better robustness than larger ViT-B/16”, the even larger model Vit-L/16 also shows better robustness than ViT-B/16, and is on par or even more robust than ViT-S/16, which contradicts the explanations in the paper.
>
> **A7**: Our original statement says: “[10] mentioned that a larger model does not necessarily imply better robustness. It can be confirmed by our experiments where ViT-S/16 shows better robustness than larger ViT-B/16 under both PGD attack and AutoAttack.” In this case, simply adding transformer blocks to the classifier cannot guarantee better robustness.” We refer to this result as a validation of the previous works’ observation, that simply increasing the model size of a CNN does not necessarily boost the adversarial robustness, also applies to “ViTs with varying transformer blocks” in our experiments. We have refined our statement in the revised version.
>
> **Q8**: The “low-level explanation” remains only a possible explanation without a definite answer, a proof, or more convincing experiments to support this explanation. The link between frequency, robustness, and low-level features is not clear to me. I hope the authors could provide a proof, an extended discussion with supporting citations, or an extensive supportive experimental evaluation.
>
> **A8**: We refer to [1, CVPR 2020] "High-frequency component helps explain the generalization of convolutional neural networks." as a supporting reference. As discussed in [1], CNNs learn and rely on some high-frequency features to make predictions. However, these features only have superious correlation with labels and do not have semantic meanings as humans define partially causing the existence of adversarial examples. And we show in our frequency study experiments (Section 6.1 of the original submission and Section 5.1.2 of the revised version) that ViTs rely less on high-frequency features compared to CNNs and possess better adversarial robustness especially against high-frequency perturbations. So frequency is one way to explain robustness, by showing that ViTs capture less high-frequency but non-robust patterns than CNNs in comparison. We also have other evidence like certified robustness in Section 5.3, which shows provable guarantees on the robustness gain of ViTs over CNNs.
>
> [1] Wang, Haohan, et al. "High-frequency component helps explain the generalization of convolutional neural networks." CVPR 2020.
>
> [2] [Naseer et al. (2021b)] On Improving Adversarial Transferability of Vision Transformers
>
> [3] [A] Towards Transferable Adversarial Attacks on Vision Transformers; AAAI 2022
>
> [4] [Salman et al. (2021)] Certified patch robustness via smoothed vision transformers
>
> [5] [Paul & Chen (2021)] Vision Transformers are Robust Learners
>
> [6] [Mahmood et al. (2021)] On the Robustness of Vision Transformers to Adversarial Examples
>
> [7] [D] Adversarial Robustness Comparison of Vision Transformer and MLP-Mixer to CNNs; BMVC 2021
>
> [8] [B] Pyramid Adversarial Training Improves ViT Performance; ArXiv 2021
>
> [9] [C] Towards Robust Vision Transformer; CVPR 2022
>
> [10] Using pre-training can improve model robustness and uncertainty; ICML 2019

---

### Review · Reviewer_Z4jg · 2022-07-15

**Summary Of Contributions:**

The proposed work conducts an extensive study on the difference between adversarial robustness of Vision transformer (ViT) and CNN architectures. Authors consider different networks and observe that transformers are more robust than CNNs for different attack settings such as white-box, transfer and certified robustness. Authors also show that adversarial training also benefit transformers achieving better robustness than CNNs. A frequency analysis is also performed where it is shown that ViTs contain less low-level information, thereby contributing to the robustness.

**Requested Changes:**

— A comparable setting similar to [1,2] above can help understand what aspects of vision transformer contribute to the robustness. A discussion on the difference in inferences drawn compared to [3,4] can also be included in the frequency study section.

— Robustness transfer for different computer vision tasks such as object detection can also be considered for evaluation.

— Authors show that Sharpness Aware Minimization (SAM) can improve robustness significantly. It would be interesting to see if this can be applied to other architectures as well.

Minor:

—  Page 10, reference missing to appendix

**Strengths And Weaknesses:**

**Strengths:**

— A comprehensive study is conducted across different architectures and comparison is performed. Such a study can benefit the community in understanding the properties of these architectures.

— Different attacks are considered to understand the robustness of transformer architectures.

— The Frequency study helps in understanding the type of features learned by transformer architectures.

**Weaknesses:**

— Although a thorough evaluation is conducted across architectures, it raises a question whether these are fair evaluations. [1] show that vision transformers might appear more robust than CNNs, but they need to account for the different training strategies and the architectural details such as the activation functions used (ReLU vs GeLU). With comparable training strategies, CNNs are shown to be as robust as ViTs on different attacks. Hence, the comparisons shown in this paper might not be a fair reflection on the whether robustness stems from the details described in [1,2] or due to other factors.

— Authors conduct a frequency study and show that CNNs are more sensitive to high-frequency perturbations than transformers, thereby contributing to difference in robustness. However, previous works [3,4] have already studied this for CNNs. Hence, novelty and contributions of the proposed work remains limited.

[1] - Are Transformers More Robust Than CNNs?,arXiv:2111.05464

[2] - Can CNNs Be More Robust Than Transformers? , arXiv:2206.03452

[3] -  High Frequency Component Helps Explain the Generalization of Convolutional Neural Networks, arXiv:1905.13545

[4] - Dissecting the High-Frequency Bias in Convolutional Neural Networks, CVPR 2021 workshop

---

> ### Author Response · Authors · 2022-08-03
> **Response to Reviewer Z4jg (1/N)**
>
> We thank the reviewer's comments and have made the following changes:
>
> **Revised terms**:
>
> * We added a discussion about different experimental settings across existing works. We also make a note that we follow [3]’s setting to compare publicly available neural network models that were well-tuned separately. We assume there is no definite “fair” experimental setting as any specific training scheme could be biased to some architectures.
>
> * We reorganized our experiment and reasoning sections (Section 5 and Section 6 in the original submission, now merged in Section 5 in the revised version) to highlight our unique contributions and special observations. We also put the reasoning and observations following corresponding experimental results directly to make them easier to follow.
>
> * We added a discussion about our novelty compared to previous frequency study on CNNs in Section 5.1, highlighting our work is distinguished from them by focusing on the sensitivity to different frequency-filtered adversarial attacks, while previous works mainly focus on clean input images. And we include ViTs and modern CNNs in our experiments, and find ViTs have natural resistance to high-frequency adversarial perturbations.
>
> * As we only focus on the image classification tasks, we included new references in Section 2 that study the object detection and semantic segmentation tasks as related works. We also provide analysis about why SAM helps improve models’ robustness and leave it to future work to check if this also applies to CNNs due to time limit. We also plan to opensource our work so that future studies can benefit from our findings.
>
> * We also discussed the effect of training strategies and architecture details in Section 5.3, showing modern CNN designs that borrow similar design principles from ViTs, (e.g, the activation function, larger kernel, layer norm, etc.), can help bridge the performance gap between ViTs and CNNs not only in terms of clean accuracy, but also certified and empirical robust accuracy.
>
> **Responses to the comments**:
>
> **Q1**: Fair experimental setting.
>
> **A1**: We understand the reviewer’s concern and we agree that experimental setting could be one of the causes to bring better adversarial robustness to vision transformers. However, there are many ways to define a “fair” experimental setting in studying adversarial robustness. The experimental setting we adopted follows [3], which takes different neural network models trained and tuned separately rather than using a common but potentially suboptimal setting (the latter could be biased to some models as well). Therefore, our analysis is based on public available model checkpoints that are well tuned and widely used in the literature. And such analysis could be easily extended to new structures and new training techniques. We added a discussion about this at the beginning of Section 3 (Model Architectures) to note the different settings used for comparison across works.
>
> Besides, we found introducing CNN blocks to the vision transformer with all training procedures unchanged will decrease the adversarial robustness which is not studied in [1]. [2] further went into details about which components contribute to the robustness of vision transformers and debated with [1] on the generalization ability of vision transformers.
>
> Moreover, what is unique to our study is that we also provide certified robustness analysis using denoised smoothing in Section 5.3, which would weaken the effect of different data augmentation skills used in the training and show the provable adversarial robustness comparison after denoising. Also, shifting focus to certified robustness instead of empirical robustness may also provide a more meaningful argument about the differences in adversarial robustness. We supplemented more results about the certified robustness analysis in Section 5.3 of the revised version.
>
> [1] Are Transformers More Robust Than CNNs?,arXiv:2111.05464
>
> [2] Can CNNs Be More Robust Than Transformers? , arXiv:2206.03452
>
> [3] Vision Transformers are Robust Learners. AAAI 2022

---

> ### Author Response · Authors · 2022-08-03
> **Response to Reviewer Z4jg (1/N, N=2)**
>
> **Q2**: Novelty of frequency study.
>
> **A2**: Previous works [1,2] conducted frequency analysis on CNNs, but they mainly focus on clean accuracy instead of adversarial robust accuracy, not even adversarial robustness comparison across different model architectures. [1] shows there exit high-frequency patterns in the clean image inputs that have high correlation with labels but little semantic information. Models picking up such patterns for classification tend to have lower clean accuracy on the test set.  [2] similarly studies the importance of different frequency-patterns to models’ classification. Our work distinguishes from these works by directly looking into the adversarial examples in the frequency domain. Besides, the prior works only look at the CNNs, and it remains unclear about the frequency property of new architectures (ViTs and MLP-Mixer etc.). And most importantly, we found ViTs have less bias towards those high-frequency patterns that have spurious correlation. We attribute ViTs’ natural resistance to such tendencies that appear for traditional CNNs. Therefore, the contributions of the references given by the reviewer are not similar to our work, instead, the findings of these works inspire and support our idea. We have included these references and relative discussions in Section 5.1 of our revised version.
>
> [1] High Frequency Component Helps Explain the Generalization of Convolutional Neural Networks.
>
> [2] Dissecting the High-Frequency Bias in Convolutional Neural Networks.

---

### Review · Reviewer_BF1i · 2022-07-16

**Summary Of Contributions:**

This paper addresses adversarial robustness for vision transformers, especially on the classification tasks. Authors test a few white-box attacks and transfer-based black-box attacks on CNNs, transformers and their combinations. Through experiments, authors claim that vision transformers are usually more robust to adversarial attack, and adding more CNN components into transformers will degrade the robustness. Authors claim that transformers focus less on low level pattern and verifies the claim with experiments and visualizations.

**Broader Impact Concerns:**

No broader impact concerns regarding the submission.

**Requested Changes:**

Some nitpickings:
* Page 4 line 5: should $x_i\in\mathbb{R}^{N \times (P^2 \cdot C\)}$ or $x_i\in\mathbb{R}^{P^2\cdot C}$? It seems that $x_i$ is only one patch (e.g. 16 by 16 one).
* Page 5 line 4: what is the subscript $i$ in $x^{adv}_{t+1,t}$? I don't see $i$ in Eq 1 or Eq 2.
* Page 10 last paragraph: "Figure 3 in Appendix ??". It should be in the main paper?
* Figure 2 seems to be too big and doesn't contain useful information. Section 3.1 already covers everything.

**Strengths And Weaknesses:**

Strengths:
+ This paper companies a broad range of vision models, including CNNs, transformers and their combinations, on the same settings.
+ It's interesting to see that the adversarial examples produced from ViT can transfer well to CNNs.

Weaknesses:
- Most adversarial attacks are PGD's and something similar. Other types of attacks are missing, such as optimization based attack or query based attack. It is unclear whether the current findings would also hold on those types of adversarial attacks.
- The attack radius used for the experiments are quite small, and most results are reported with $\epsilon$ less than or equal to 0.01. According Figure 8 in Appendix B, growing $\epsilon$ from 0.01 to 0.1 will lead to a huge attack success rate increase. It would be interesting to provide a denser result curve wrt $\epsilon$, and analyze whether the trend aligns with the low-frequency and high-frequency story. Also in Figure 7 of Appendix B, it would be better to show the accuracy or ASR wrt number of iterations -- the cross entropy loss metric does not align with other experiment results reported in the paper.
- In Figure 4, can we add more analysis about why ViT adversarial examples are better transferred to CNNs? When $\epsilon$ is 0.1, their adversarial examples are even stronger than the CNN counterparts. But when $\epsilon$ goes lower to 0.01, the transferability goes bad. Does that relate to the conjecture authors provide regarding why transformers are more robust?
- In Table 5, when adding more transformer blocks to T2T-ViT, the low-pass attack rate also goes down. Any explanation to that?

---

> ### Author Response · Authors · 2022-08-03
> **Response to Reviewer BF1i (1/N)**
>
> **Revised terms**:
> * We modified the illustration of different attacks we use in Section 4 (Adversarial Robustness Evaluation Methods) in the revised version. We clarify the adversarial attacks we use for evaluation including PGD, AutoAttack(an ensemble of two auto PGD attacks, an optimization-based attack, and a query based black-box attack), and transfer-based black box attack, together with our crafted frequency-filtered PGD attack and denoised randomized smoothing used in the frequency and certified robustness analysis. We also re-arranged the experiment section according to this clearer clarification to avoid confusion.
> * We added a discussion on the observation that ViT adversarial examples are better transferred to CNNs in Section 5.2.2 in the revised version.
> * We also revised the typos as mentioned in the first and fourth requested changes, and deleted the original Figure 2 as suggested. Regarding the second point of requested change, the i indicates the index of the sample as we noted $(x_i, y_i)\in \mathcal{D}$. The equations 1 and 2 are for a single sample which equals $x_i,\ \forall i$.
> * We also re-organized the experiment section with discussions directly following their corresponding experimental results, and we highlighted our key observations with numbers to facilitate reading, e.g. the discussion about adding transformer blocks to T2T-ViT could lead to improvement wrt robust accuracy.
> * We added the results of robust accuracy comparison versus varying PGD attack steps in Figure 6 of Appendix A in the revised version. The robust accuracy results are consistent with the conclusion we drew from the loss curve in the original submission.
>
> **Responses to the comments**:
>
> **Q1**: Most adversarial attacks are PGD's and something similar. Other types of attacks are missing, such as optimization based attack or query based attack. It is unclear whether the current findings would also hold on those types of adversarial attacks.
>
> **A1**: We understand the reviewer’s concern. In fact, In Table 3 in Sec 5.1 of the original submission (Table 4 in Sec 5.2.1 of the revised version), we also reported the attack results based on the AutoAttack (AA), which is a combination of gradient-based white box attack (APGD-CE, APGD-DLR), optimization based attack (FAB) and query based black-box attack (Square Attack). We have revised the illustration of different attacks we use in the experiments in Section 4 of the revised version for clearer clarification.
>
> **Q2**: The attack radius used for the experiments are quite small, and most results are reported with less than or equal to 0.01. According Figure 8 in Appendix B, growing from 0.01 to 0.1 will lead to a huge attack success rate increase. It would be interesting to provide a denser result curve wrt, and analyze whether the trend aligns with the low-frequency and high-frequency story. Also in Figure 7 of Appendix B, it would be better to show the accuracy or ASR wrt number of iterations -- the cross entropy loss metric does not align with other experiment results reported in the paper.
>
> **A2**: We itemize our responses as follows.
> * The experiments are conducted on the ImageNet dataset where prior works [CITE] often use the perturbation budget of 1/255 or 2/255 for evaluation, so 0.01 is not so small considering the dataset. Moreover, we focus on the adversarial robustness of naturally trained models. Without adversarial training, it is difficult for models to be robust against large perturbations.
> * As shown in the Table 2 of the revised version, CNNs have moderate robust accuracy drop wrt low-frequency perturbations but severe decrease against high-frequency perturbations when increasing the perturbation rates. For example, when increasing the attack radius from 0.001 to 0.01, the robust accuracy of ResNet50-32x4d decreases from 75.0% to 59.0% (by 16.0%) for low-pass adversarial attack, but 47.0% to 3.3% (by 43.7%) for high-pass adversarial attack, indicating the high-frequency perturbations are the main cause to the low robust accuracy for CNNs. However, when increasing the attack radius from 0.001 to 0.01, ViT-B/16 decreases from 71.9% to 55.8% (by 16.1%) for low-pass adversarial attack, but only 66.3% to 33.4% (by 32.9%) for high-frequency-perturbations, showing much better resistance to high-frequency perturbations compared with CNNs. This is consistent with our previous analysis and we have added the above discussion in Section 5.2.2 of the revised version.
> * We added the results of robust accuracy comparison versus varying PGD attack steps in Figure 6 of Appendix A in the revised version. The robust accuracy results are consistent with the conclusion we drew from the loss curve in the original submission.

---

> ### Author Response · Authors · 2022-08-03
> **Response to Reviewer BF1i (2/N, N=2)**
>
> **Q3**: In Figure 4, can we add more analysis about why ViT adversarial examples are better transferred to CNNs? When eps is 0.1, their adversarial examples are even stronger than the CNN counterparts. But when eps goes lower to 0.01, the transferability goes bad. Does that relate to the conjecture authors provide regarding why transformers are more robust?
>
> **A3**: Thanks for pointing out this phenomena which we think also attributes to the superior adversarial robustness of vision transformers to the high-frequency patterns. As studied in [1], they found and called those features less prone to be attacked by adversarial attacks as “robust features”, which is consistent with our understanding of the “high-level” features that are preferable for neural networks to learn. And those noise-like “non-robust” features are similar to our high-frequency features that contain low-level information. Since the “non-robust” high-frequency patterns are generated with specific models, adversarial perturbations with regard to such patterns should be assumed to be harder to transfer. Since ViTs are less sensitive to such high-frequency patterns, we assume the adversarial perturbations will be forced to rely less on model-specific feature patterns and thus be easier to transfer. It is interesting to notice that ViTs’ adversarial examples could be even stronger than the CNN counterparts with large attack radii, and we think it’s because the transferable adversarial perturbations need larger attack radii which could be further explored. We have added this discussion in Section 5.2.5 of the revised version.
>
> [1] Adversarial Examples Are Not Bugs, They Are Features
>
> **Q4**: In Table 5, when adding more transformer blocks to T2T-ViT, the low-pass attack rate also goes down. Any explanation to that?
>
> **A4**: As we explained in the original Section 6, the tokens-to-token block in T2T-ViT helps the model to learn more information about the high-frequency patterns that have high correlation with labels but little semantic information, which benefits the ViTs to achieve comparable performance when trained from scratch without pre-training but at the cost of sacrificing the original adversarial robustness of ViT. To improve our presentation, we re-organized the experiment section with discussions directly following their corresponding experimental results in the revised version. We also highlight our key observations with numbers to facilitate reading.

---

### Public Comment · ~Kishaan_Jeeveswaran1 · 2022-06-13
**Reference Suggestion**

Dear Authors,

Congratulations on the submitted paper at TMLR. We find your paper insightful and interesting. In our accepted paper at VISAPP conference, we performed a comprehensive study comparing Vision Transformers and CNNs on various robustness benchmarks including adversarial attacks in object detection and segmentation. In addition, we provided some insights on why Transformers are in general robust compared to CNNs. We find our paper very relevant to yours and it would be appreciated if you could discuss it in your paper.


Thank you and best wishes!

Best regards,

Kishaan


@conference{visapp22,
author={Kishaan Jeeveswaran. and Senthilkumar Kathiresan. and Arnav Varma. and Omar Magdy. and Bahram Zonooz. and Elahe Arani.},
title={A Comprehensive Study of Vision Transformers on Dense Prediction Tasks},
booktitle={Proceedings of the 17th International Joint Conference on Computer Vision, Imaging and Computer Graphics Theory and Applications - Volume 4: VISAPP,},
year={2022},
pages={213-223},
publisher={SciTePress},
organization={INSTICC},
doi={10.5220/0010917800003124},
isbn={978-989-758-555-5},
issn={2184-4321},
}

---

> ### Author Response · Authors · 2022-06-14
> **Thank you for your reference**
>
> Dear Kishaan,
>
> Thank you for pointing us to your work. We will cite and discuss this work in our revised version.
>
> Authors

---

### Decision · Action_Editors · 2022-09-28

**Recommendation:** Accept as is

**Comment:**

The paper received reviews from four expert reviewers in the field. Both Reviewer Z4jg and Reviewer N2u2 are satisfied with the author's paper revision and recommend to accept. Reviewer BF1i did not enter the final recommendation, but the AE reads the comments and the authors' revision and believes that there are no outstanding critical concerns. Reviewer UVpe suggested that most of their technical concerns have been sufficiently address. There was a remaining concern regarding the novelty with respect to several concurrent work (posted on arXiv). As 1) novelty is not one of TMLR's acceptance criteria and 2) arXiv publication timelines are irrelevant and should be ignored, the AE does not find this concern a valid ground for diminishing the contributions of this work.

Considering all the factors, the AE agrees with the reviewers that extensive analysis of adversarial robustness for vision transformers provides valuable insights to the community. The authors' revision also significantly improves the clarity of the exposition. The AE thus recommends to accept as is.